# Unpacking Evaluation Pitfalls on Standard GNN Benchmarks

## Abstract

Graph Neural Networks (GNNs) have achieved substantial progress in graph-structured learning, with recent innovations targeting heterophilic graphs and attention-based designs such as Graph Transformers. These models are typically evaluated on widely used standard benchmark datasets for node and graph classification. In this work, we identify a critical and often overlooked issue: these widely used benchmarks frequently suffer from significant class imbalance. Despite this prevalence, the GNN community predominantly relies on individual aggregate metrics namely *standard accuracy* and *AUROC*, on these datasets, often overlooking their limitations and target utility. While convenient, the existing aggregate measures could obscure class-level disparities and lead to incorrect conclusions about architectural effectiveness. Our work provides empirical evidence to demonstrate this limitation and advocate for a more robust evaluation framework that incorporates a diverse set of metrics (including balanced accuracy, AUPRC, and per-class metrics) to enable a transparent and reliable assessment of GNN capabilities.

## 1 Introduction

Graph Neural Networks (GNNs) have emerged as a powerful paradigm for analyzing graph-structured data, achieving state-of-the-art results across a multitude of tasks including node classification, link prediction, and graph classification (Veličković et al., 2017; Hamilton et al., 2017; Kipf, 2016). The versatility of GNNs has led to their widespread application in various fields, including bioinformatics, social network analysis, chip design, recommendation systems, etc. (Wu et al., 2020; Sharma et al., 2024; Zhang et al., 2021). This rapid advancement is largely driven by the continuous innovation in GNN architectures, and researchers continue to propose sophisticated GNN models to tackle challenges such as heterophily and long-range dependencies. This includes specialized heterophily GNNs (Zhu et al., 2020; Li et al., 2022; Maurya et al., 2021; Zhu et al., 2021), meticulously designed for graphs where connected nodes often belong to different classes, as well as advanced designs like graph Transformers and other attention-based mechanisms that aim to capture intricate long-range dependencies (Ying et al., 2021; Veličković et al., 2017; Dwivedi & Bresson, 2020; Rampášek et al., 2022; Shirzad et al., 2024; Kong et al., 2023).

As the field matures, evaluation protocols have standardized around widely adopted benchmark datasets such as Questions, Squirrel-filtered, Amazon-ratings, and ogbn-arxiv (Platonov et al., 2023b; Hu et al., 2020) for node classification, alongside datasets such as ogbg-molhiv (Hu et al., 2020), COLLAB (Rossi & Ahmed, 2015), and COX2 Sutherland et al. (2003) for graph classification. The rigorous evaluation of these cutting-edge models on a diverse and widely adopted set of benchmark datasets is crucial for the field's progression, as it provides a common ground for fair comparison, quantifies the effectiveness of new approaches, highlights areas for improvement, and ultimately fosters further innovation by setting clear targets for research and development (Platonov et al., 2023b; Bechler-Speicher et al.; Luo et al., 2024; 2025b).

In our study, we first highlight a critical yet often overlooked property of several of these GNN benchmark datasets: the *presence of significant class imbalance*. Although class imbalance is a fundamental property of many real-world datasets, a key finding from our investigation is that the vast majority of GNN research continues to use mainly accuracy and area under receiver operating characteristics(AUROC) (Bradley, 1997) for evaluating model performance on these standard

benchmarks, which are imbalanced. Although these metrics are standard and have their own benefits, they may not be the ideal choice for all data sets. The utility of these metrics(or even other metrics such as area under precision-recall curve-AUPRC, balanced accuracy (Ghanem et al., 2023; Seo et al., 2021)) is highly dependent on the specific context and the problem at hand (Josephine, 2017; Ghanem et al., 2023; Owusu-Adjei et al., 2023; Saito & Rehmsmeier, 2015; Akosa, 2017; Krawczyk, 2016; Johnson & Khoshgoftaar, 2019; Seo et al., 2021). Crucially, in our investigation, we found the majority of the GNN research works in which *accuracy*, a metric particularly insensitive to class imbalance (Guesné et al., 2024; Thölke et al., 2023), was extensively reported even when the underlying widely used standard benchmarks such as *Amazon-ratings* (Platonov et al., 2023b), *ogbn-arxiv* (Hu et al., 2020), and *Squirrel-filtered* (Platonov et al., 2023b) are severely imbalanced. Several GNN models have reported significant improvements on these standard benchmarks with the above metrics, and consequently assert architectural superiority Luo et al. (2024). Furthermore, we also highlight that beyond research works involving architectural innovations, we also observe that benchmarking studies have overlooked this while evaluating on these standard GNN benchmark datasets(Luo et al., 2025b; Park et al., 2025b; Platonov et al., 2023a). The current evaluation protocols on these benchmarks may provide an incomplete understanding of a model's true capabilities (Guesné et al., 2024; Thölke et al., 2023). To ensure a robust and truthful evaluation, it is crucial to move beyond single metrics. This work aims to redirect the field toward evaluation protocols that capture various aspects of model performance, promoting a more robust evaluation of GNNs on widely used benchmarks.

In support of these crucial observations and the significant relevance of the above works, our work dives into the evaluation aspect of GNNs on several standard benchmark datasets. Towards this, we make the following contributions.

1. We demonstrate the prevalence and significant impact of class imbalance on standard GNN benchmarks, revealing that existing literature often relies on metrics insensitive to this issue.

2. We empirically show how relying solely on aggregate metrics like accuracy and AUROC on these standard GNN benchmarks could lead to incomplete performance assessment of several GNN models, especially for minority classes in both node classification and graph classification.

3. We propose and advocate for a more robust and diverse evaluation framework for GNN research. This framework also emphasizes including metrics that provide additional insights into performance, including per-class metrics such as class-level F1-scores, aggregate AUPRC and balanced accuracy, as these can reveal performance nuances that existing aggregate measures alone might overlook. The motivation is to ultimately foster more transparent and reliable progress in the field.

## 2 RELATED WORK

**Neural Networks for Graphs:**   Graph Neural Networks (GNNs) have emerged as powerful class of models for tasks such as node classification, graph classification, link predictione etc. by iteratively aggregating information from a node's local neighbors, combining graph structure and features to learn informative representations (Kipf, 2016; Veličković et al., 2017). Despite their success, GNNs could face challenges such as over-smoothing, over-squashing, limited ability to handle some types of heterophily. To address these limitations, specialized neural architectures have been proposed for heterophilous graphs (Maurya et al., 2021; Zhu et al., 2021; Pirro; Li et al., 2022). More recently, Graph Transformers (GTs) have also been proposed which use self-attention to capture global interactions between any nodes in the graph (Deng et al., 2024; Chen et al., 2024b; Ma et al., 2024; Rampášek et al., 2022; Ying et al., 2021), with several studies demonstrating their impact on graph-level tasks such as molecular property prediction.

**Benchmarking GNNs for Graph Problems:**   Recently, impactful studies, including the work by (Platonov et al., 2023b; Luo et al., 2024) studied GNNs under heterophily. These studies demonstrated that GNN architectures can match specialized heterophily-focused as well as Graph transformer models when evaluated on carefully curated, bias-free heterophilous datasets. Similarly, Luo et al. (2025b) demonstrated that classic GNNs in many cases can match the performance of Graph

transformers for graph classification tasks. These findings highlight the importance of rigorous benchmarking in assessing the architectural effectiveness of different methods. Motivated by these observations, our work further emphasizes the importance of using appropriate evaluation metrics on standard GNN benchmarks, especially in the presence of data imbalance, to better understand and assess GNN performance across diverse conditions.

**Current evaluation practices on standard GNN benchmarks**   We surveyed several research works which performed evaluation on the standard benchmarks namely Amazon-ratings, Questions, Squirrel-filtered, ogbn-arxiv, COLLAB, COX2, and ogbg-molhiv. We observe that the vast majority of works on graph learning which is focused on designing GNNs, use standard accuracy as metric on multi-class datasets (e.g., Amazon-ratings, Squirrel-filtered, and COLLAB), while AUROC is the widely used standard metric for binary tasks such as Questions, ogbg-molhiv, and COX2  Zhao et al. (2023); Zhou et al. (2023); Platonov et al. (2023a); Luo et al. (2025b; 2024). A full list of references is provided in Appendix A.1 for space considerations.

# 3  INVESTIGATING ISSUES WITH CURRENT EVALUATION ON STANDARD GNN BENCHMARK DATASETS.

To comprehensively investigate the issues with current GNN evaluation on standard GNN benchmarks, we consider a diverse set of widely used datasets, categorized by their primary task: node classification and graph classification. For each dataset, we first analyze its class distribution and discuss its implications for commonly used evaluation metrics. We then analyze the performance of various neural models on these datasets using both imbalance-aware and imbalance-insensitive metrics.

Table 1: Statistics of Benchmark Node Classification Datasets

| Statistic | Amazon-ratings | Questions | Squirrel-filtered | ogbn-arxiv |
|---|---|---|---|---|
| Nodes | 24492 | 48921 | 2223 | 169343 |
| Edges | 93050 | 153540 | 46998 | 1166243 |
| Node features | 300 | 301 | 2089 | 128 |
| Classes | 5 | 2 | 5 | 40 |
| Class Ratio | $C_0$: 0.267
$C_1$: 0.367
$C_2$: 0.231
$C_3$: 0.08
$C_4$: 0.04 | $C_0$: 0.97
$C_1$: 0.03 | $C_0$: 0.340
$C_1$: 0.232
$C_2$: 0.178
$C_3$: 0.144
$C_4$: 0.104 | $C_{16}$: 0.161
$C_{24}$: 0.131
$C_{28}$: 0.126
...
$C_{12}$: 0.0002
$C_{35}$: 0.00075
$C_{21}$: 0.0023 |

Table 2: Statistics of Graph Classification Datasets

| Statistic | ogbg-molhiv | COX2 | COLLAB |
|---|---|---|---|
| Number of Graphs | 41127 | 1238 | 5000 |
| Avg Nodes per Graph | 25.5 | 25.9 | 74.49 |
| Avg Edges per Graph | 27.5 | 27.9 | 2457.78 |
| Number of Classes | 2 | 2 | 3 |
| Class Ratio | $C_0$: 0.036
$C_1$: 0.963 | $C_0$: 0.782
$C_1$: 0.218 | $C_0$: 0.52
$C_1$: 0.325
$C_2$: 0.155 |

## 3.1 DATASETS

We analyze critical benchmarks across two domains: node classification, focusing on key heterophilic datasets widely used for assessing GNNs and Graph Transformers. For graph classification, we examine several standard benchmark datasets utilized to measure the effectiveness of different GNN architectures. The dataset statistics are present in Table 1 and 2.

**Node Classification:**

- **Questions** (Platonov et al., 2023b): In this datasets, the classification task is to predict which users remained active on the Yandex Q question-answering website. The dataset is highly imbalanced; approximately 97% of users belong to the "active" (majority) class, and 3% to the "inactive" (minority) class.

- **Squirrel-filtered** (Platonov et al., 2023b): A network of Wikipedia pages, where nodes are classified into one of five categories based on monthly traffic, reflecting popularity. Despite accuracy being the widely used metric, this dataset has significant class imbalance; for instance, Class 0 contains 756 nodes, while Class 4 has only 233. Understanding predictive performance across these different popularity levels is essential, as these groups reflect meaningful, real-world differences in page usage.

- **Amazon-ratings** (Platonov et al., 2023b): This dataset is derived from Amazon's product co-purchasing network. Nodes represent products (eg:- books, DVDs), and edges connect products frequently bought together. The objective is to predict a product's average customer rating across five classes(1-5 stars). There is a severe class imbalance, as the largest class represents more than 37% of the data, while the smallest accounts for less than 5%. Given that different classes may hold varying importance in this task, the prevalent use of accuracy could lead to misleading conclusions, particularly as minority classes ($C_3$ and $C_4$) collectively represent a small fraction of the data.

- **ogbn-arxiv** (Hu et al., 2020): The ogbn-arxiv dataset is a citation network where each node is a Computer Science arXiv paper and each edge is a citation. Every paper belongs to one of 40 subject categories, e.g., cs.AI, cs.LG, cs.OS etc.. The goal is to predict which of the categories each paper belongs to, a process that assists arXiv moderators. A critical challenge in this dataset is the severe class imbalance: the top 4 most frequent classes are disproportionately dominant, accounting for approximately 48% of the total samples, while the bottom 10 least frequent classes contain less than 2.4% of the samples combined. This imbalance makes accuracy a potentially unreliable indicator of overall model effectiveness, especially for identifying papers in niche or relatively less-represented categories. Due to space limitations, the detailed statistics of this dataset are present in App. D.

**Imbalance in Node Classification Benchmark Datasets:** The datasets for node classification in Table 1 clearly demonstrate class imbalance in several standard benchmark datasets. On imbalanced datasets like Squirrel-filtered, Amazon-ratings, and ogbn-arxiv, the common use of accuracy as an evaluation metric might not present a true picture. In an imbalanced dataset, accuracy could lean towards the majority class; it doesn't truly reflect how well a model performs overall, especially when every class holds equal importance unless specified (Guesné et al., 2024; Thölke et al., 2023). This can unintentionally give us an inaccurate view of a model's or the model configuration's real capability, and could lead to suboptimal choices.

**Graph Classification:**

- **ogbg-molhiv** (Hu et al., 2020): A prominent datasets from the popular Open Graph Benchmark (OGB) designed for molecular property prediction. This dataset presents a challenging binary graph classification task characterized by extreme class imbalance. As detailed in Table 2, the positive class (inhibitors) represents only about 4% of the data, while the negative class (non-inhibitors) makes up the remaining 96%. High-quality prediction for the minority positive class is valuable for bio-scientists to decide which molecules to advance to clinical trials. Consequently, robust evaluation hinges on metrics that are highly sensitive to the performance on this crucial minority class.

- **COLLAB**: (Rossi & Ahmed, 2015): The COLLAB dataset is comprised of graphs, each representing a researcher's ego network. In these networks, nodes signify researchers, and

edges denote co-authorship within a specific scientific field. The primary objective is to classify the scientific field (class) associated with a given subgraph. A notable characteristic of this dataset, as detailed in Table 2, is its significant class imbalance, where one class constitutes a significantly lower(15%) percentage of the total samples in the three-class classification task. The widely reported metric on this dataset is accuracy.

- **COX2** (Sutherland et al., 2003): COX2 is another widely used dataset for molecular graph classification, focusing on a critical biological target: the cyclooxygenase-2 (COX-2) enzyme. The primary task is to predict if a chemical compound can inhibit this enzyme, which is a critical target for developing anti-inflammatory drugs. The dataset shows a significant imbalance with approximately 21% of positive samples and 79% negative samples.

**Imbalance in Graph Classification Benchmark Datasets:** The datasets discussed, as detailed in their descriptions and in Table 2, exhibit significant class imbalance. The case of ogbg-molhiv with $\approx 4\%$ positive samples is particularly noteworthy. The widely used metric for this dataset is AUROC (Luo et al., 2025b; Hu et al., 2020), also proposed by the ogb source (Hu et al., 2020). This contrasts with another dataset, ogbg-molpcba, from the same ogb source, which, with its more severe imbalance of about $\approx 1.4\%$ positive samples (and multiple classes), prompts the use of Average Precision. While a $1.4\%$ rate in ogbg-molpcba is notably more imbalanced which is acknowledged by the community (Hu et al., 2020), a $4\%$ positive class rate in ogbg-molhiv is still highly skewed and far from a balanced distribution, making the choice of evaluation metric critical.

### 3.2 EXPERIMENTAL SETUP

#### 3.2.1 EVALUATION METRICS

To ensure a rigorous and comprehensive GNN performance assessment, we extend beyond the standard accuracy and AUROC. Metric selection is paramount for imbalanced classification, where existing aggregate measures, often insensitive to imbalance might not reflect the true overall picture. Consequently, our evaluation employs both existing as well as new aggregate metrics, as well as per-class metrics, to provide granular insights.

**Aggregate Metrics:** These metrics provide an overall summary of the model's performance across all classes. While widely used, some are more robust to class imbalance than others. For binary classification datasets, we report both AUROC (consistent with prior literature) and AUPRC. For multi-class classification, we utilize both standard accuracy (following existing works) and balanced accuracy.

**Per-class metrics:** To gain fine-grained insights into model behavior and identify specific strengths or weaknesses for individual classes, we report F1-score per class. The definitions of the metrics are present in App. B.

## 4 EXPERIMENTS

This section analyzes the performance of various models on standard GNN benchmarks. We specifically highlight the significant performance drop observed when using metrics that take class imbalance into account. Our experiments further demonstrate that different configurations of a model could yield similar results on standard (imbalance-insensitive) metrics, yet produce significantly divergent results when evaluated with imbalance-aware metrics.

### 4.1 EXPERIMENTAL SETUP

**Train/Val/Test Split:** For the node classification datasets, we follow the standard splits from the benchmarking paper that proposed these datasets (Platonov et al., 2023b; Hu et al., 2020). In graph classification, for COX2 and COLLAB, we perform an 80/10/10 split, and for ogbg-molhiv, we follow the standard split as per the benchmarking work of Hu et al. (2020).

**Models considered:** Our evaluation includes standard GNN architectures: GCN (Kipf, 2016), GraphSAGE (Hamilton et al., 2017), and GAT (Veličković et al., 2017), a Heterophily-Aware GNN FSGNN(alias FSGCN) (Maurya et al., 2021), and Graph Transformers (GTs) namely

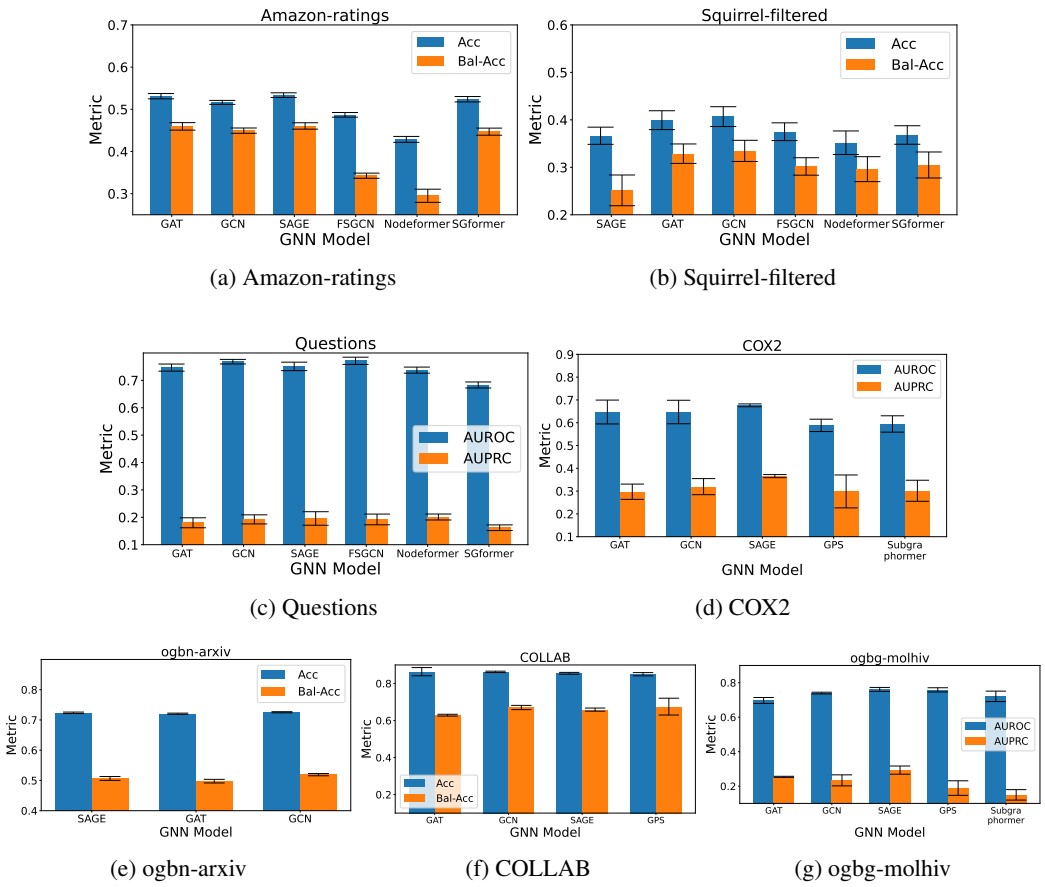

Figure 1: Comparison of performance of different models on standard GNN benchmark datasets on different evaluation metrics.

SGFormer (Wu et al., 2023) and Nodeformer (Wu et al., 2022) for node classification and GraphGPS (Rampášek et al., 2022) and Subgraphormer (Bar-Shalom et al., 2024) for graph classification. Although this list is not exhaustive, our primary objective was to assess the behavior of different models on different metrics on the standard GNN benchmarks.

**System Configuration and Parameters** We conduct our experiments on a 12GB NVIDIA GeForce GTX 1080 Ti GPU with PyTorch version 2.6.0, and PyTorch-geometric version 2.6.1. The details of hyperparameters for different models are presented in Appendix C. The code to run experiments is present at `https://anonymous.4open.science/r/eval_graph`.

## 4.2 RESULTS

**Performance variation when taking class imbalance into consideration.** In this section, we investigate how the performance of different models varies when evaluated on metrics that take into account the aspect of class imbalance. The model performance plots in Fig. 1 reveal a critical evaluation pitfall. On several imbalanced *ogbn-arxiv* node classification datasets (Fig. 1e), several models achieve a standard accuracy of $\approx 71\%$, yet the balanced accuracy metric drops sharply to $\approx 50\%$. This disparity is starkly evident from the per-class results in Fig. 1e where high-support classes like $C_{16}$ and $C_{24}$ show excellent performance (F1-score over $\approx 85\%$), while low-support classes such as $C_{21}$ and $C_{35}$ suffer significantly, with metrics falling below $5\%$. We observe that high-support classes like $C_{16}$ and $C_{28}$ show excellent performance (F1-score over $85\%$), while low-support classes such as $C_{21}$ and $C_{35}$ suffer significantly, with metrics falling below $5\%$. This finding highlights a crucial point: aggregate accuracy on these standard GNN benchmarks alone can present

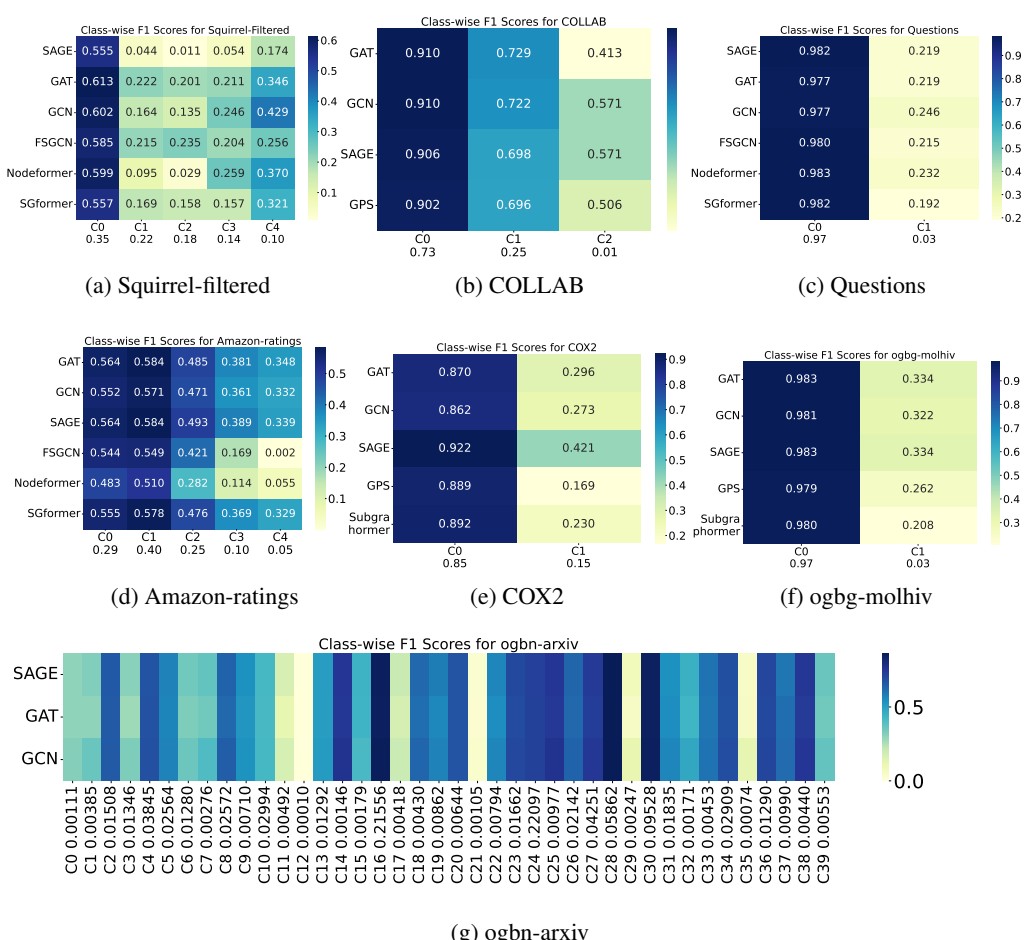

Figure 2: Classwise results on GNN Benchmarks for the aggregate results presented in Fig. 1. As evident, on several datasets, class-specific performance is highly uneven, and a widely used metric such as accuracy might mask important variations in how well the model handles different classes in these imbalanced datasets. For each class, we also present the test class ratios rounded to 2 decimal places.

a distorted view of performance when class imbalance is present. Therefore, using metrics such as balanced accuracy or per-class metrics is vital for a comprehensive evaluation of a model's true capabilities. We observe similar conclusions in other datasets such as *Squirrel-filtered* and *Amazon-ratings* in Fig. 1b, 1a and 2a, 2d. It is interesting to note that on the *Squirrel-filtered* dataset in Fig 1b, the SAGE and SGFormer models achieve similar accuracy scores, but the balanced accuracy metric differs by a margin of $\approx 5\%$, underscoring that accuracy alone might not provide a complete understanding and could lead to incorrect conclusions.

Similar observations are obtained in graph classification datasets. For instance in Fig. 1f, we observe that on the COLLAB dataset, several models obtain a high accuracy score above $\approx 84\%$, however, this contrasts sharply with low balanced accuracy $\approx 67\%$. The per-class metrics for COLLAB in Figure 2b show high-performance above $\approx 85\%$ on the majority class, however, the performance on the non-majority classes is significantly lower.

In Fig. 1g, for the binary classification *ogbg-molhiv* dataset, we observe that AUROC for different models is generally above $\approx 70\%$ while AUPRC is below 30%. The low AUPRC indicates that the models struggle to achieve high precision and recall on the minority class, a small but vital set of inhibitors. For scenarios like drug development, where the cost associated with false positives could significantly outweigh that of false negatives due to expensive clinical trials, the knowledge

of a metric such as AUPRC is helpful in validating the high precision and recall required for making critical, high-certainty decisions (McDermott et al.). Hence, studying an additional metric such as AUPRC is desired when evaluating performance on this widely benchmarked dataset.

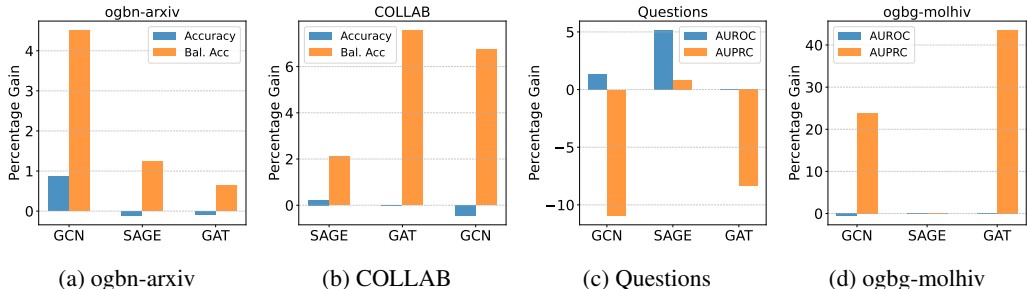

(a) ogbn-arxiv      (b) COLLAB      (c) Questions      (d) ogbg-molhiv

Figure 3: **Divergent metrics:** Different configurations of the same model can lead to similar accuracy/AUROC scores, as shown by the low percentage blue colored gain bar, but significantly different balanced accuracy/AUPRC metrics(orange bar). This highlights how a single metric can mask significant performance differences, especially in imbalanced datasets.

**Divergent Metrics: A More Nuanced Perspective on Model Performance** This section highlights the critical impact of evaluation metrics, particularly in scenarios where models achieve similar scores on standard measures like accuracy or AUROC but vary significantly on class-imbalance-sensitive metrics such as balanced accuracy and AUPRC. To illustrate this, we experimented with two model configurations(number of layers/pooling mechanism) for each model. We then analyzed the percentage change observed for each model w.r.t accuracy and balanced accuracy for multi-class datasets, and AUROC and AUPRC for binary classification datasets.

Figure 3 presents these results across different datasets. For each model, a pair of bars(blue and orange) represents the two configurations. The blue bars indicate the percentage gain of the first configuration over the second on the blue metric, while the orange bars show the gain on the orange metric. We observed, for instance, that in the Questions dataset (Fig. 3c), two GAT models yielded highly similar AUROC scores, yet their AUPRC differed by approximately 9%. Similarly, for ogbg-molhiv, the percentage gain in AUROC for two GCN models was negligible, but their AUPRC scores diverged significantly. Similar trends were found in ogbn-arxiv and COLLAB. These findings underscore that relying solely on metrics insensitive to class imbalance risks overlooking crucial performance differences. Therefore, a comprehensive understanding of diverse metrics is essential for accurately assessing a model's true capabilities and making informed selections based on the specific requirements of a given use case. For reference, absolute numeric values for this plot and model configurations are present in App. E.

**Optimizing for different objectives: Impact of validation metric** In this section, we study the impact of using different validation metrics for choosing the best model and validation epoch. In fig. 4, we observe that models selected based upon validation balanced-accuracy yield a higher performance on balanced-accuracy, highlighting the choice of validation metric also plays a role in evaluation. For example, on *Squirrel-filtered*, we observe a gain of ≈ 2−3%, on *ogbn-arxiv* ≈ 1.5−2%, and approximately 5−6% gain on COLLAB. Similarly, for binary datasets like Questions, validating with AUPRC led to a higher test AUPRC score. This additional study shows that, depending on the target use case and the relevant metric under consideration, the selected validation metric could also lead to a change in the performance.

## 5 DISCUSSION AND CONCLUSION

Our study highlights a critical and overlooked pitfall in GNN evaluation on several standard GNN Benchmarks, which are commonly used to assess the performance of GNNs: the widespread reliance on aggregate metrics like accuracy and AUROC often masks significant performance deficiencies, particularly for minority classes in imbalanced datasets. While these single metrics offer appealing simplicity, our findings unequivocally demonstrate that this convenience comes at a steep cost: a

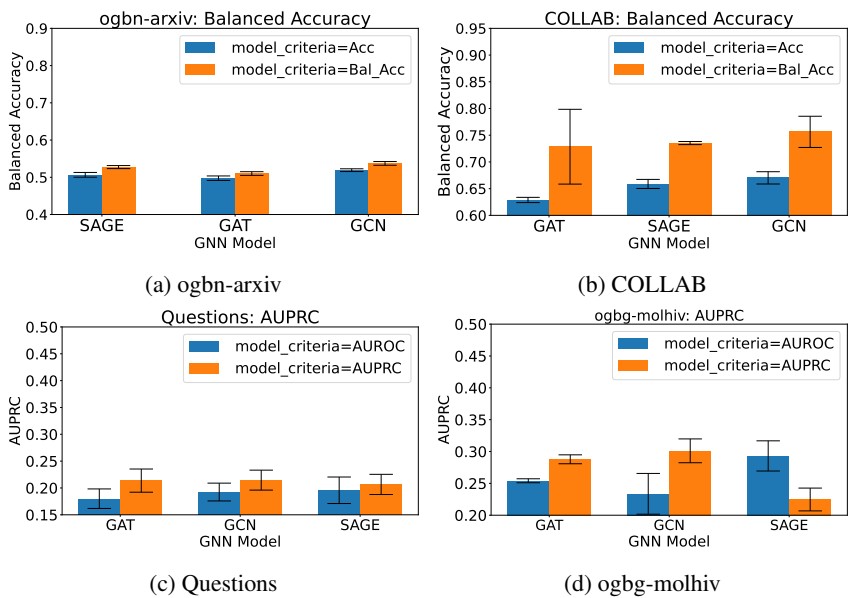

(a) ogbn-arxiv

(b) COLLAB

(c) Questions

(d) ogbg-molhiv

Figure 4: Model selection based upon different validation metrics. In this experiment, for ogbn-arxiv and COLLAB, we compared performance when accuracy and balanced accuracy are selected as validation criteria. For Questions and ogbg-molhiv, we compared with AUROC and AUPRC as validation criteria.

hidden issue of masked performance disparities. Models often achieve high scores on traditional metrics yet fail substantially when assessed with imbalance-aware measures such as balanced accuracy and per-class F1-scores. This gap is not merely a statistical nuance; it profoundly limits our understanding of a model's true generalization capabilities and its reliability in real-world, high-stakes applications like drug discovery or anomaly detection. In several cases, we also observed that model configurations could be deemed equivalent by standard metrics, but still could exhibit dramatically different performance on imbalance-aware measures. This means that selecting GNNs based solely on traditional metrics risks choosing suboptimal models.

Through a detailed analysis spanning various benchmark datasets and GNN architectures, we advocate for a fundamental shift in evaluation practices. We urge the community to adopt a comprehensive, context-aware approach utilizing diverse metrics. By moving beyond the simplicity of existing widely used aggregate scores on these standard benchmarks, researchers can gain a more accurate understanding of GNN performance, fostering the development and selection of more robust, equitable, and impactful GNN applications.

## 6 ETHICS STATEMENT

To the best of our understanding, our work does not present any new ethical concerns. All experiments are conducted on publicly available benchmark data, posing no risk to human subjects or their privacy.

## 7 REPRODUCIBILITY STATEMENT

In the anonymous repository `https://anonymous.4open.science/r/eval_graph`, we attach the code to run the experiments for this work. The experimental Sec. 4 contains information on the splits used, system configuration details, and App. C reports the parameters studied.

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

## A  APPENDIX

### A.1  RESEARCH WORKS REPORTING ACCURACY AND AUROC ON STANDARD GNN BENCHMARKS

With respect to evaluation metrics, most studies report Accuracy on multi-class datasets (e.g., Amazon-ratings, Squirrel-filtered, ognb-arxiv, COLLAB), whereas for binary classification datasets such as Questions, ogbg-molhiv, COX2, the convention is to report ROC-AUC (Zhao et al., 2023; Zhou et al., 2023; Platonov et al., 2023a; Behrouz & Hashemi, 2024; Rossi et al., 2024; Müller et al., 2023; Luan et al., 2024a; Chen et al., 2024d; Lee et al., 2023; Wan et al., 2024; Maskey et al., 2023; Gosch et al., 2023; Deng et al., 2024; Shirzad et al., 2024; Chen et al., 2024a; Zhou et al., 2023; Platonov et al., 2023a; Zhao et al., 2023; Luo et al., 2024; Yu et al., 2025; Dong & Kluger, 2023; Liao et al., 2023; Koke & Cremers, 2023; Li et al., 2024d; Wang et al., 2024a; Lu et al., 2024; Xu et al., 2024a; Li et al., 2024c; Liang et al., 2024; He et al., 2024; Ma et al., 2024; Duan et al., 2024; Zhuo et al., 2024; Attali et al., 2024; Wang et al., 2024b; Bechler-Speicher et al., 2024; Zhao et al., 2024b; Kiani et al., 2024; Gong et al., 2023; Zhou et al., 2024b; He et al., 2025; Cavallo et al., 2023; Li et al., 2025c; Sun et al., 2024a;c; Dong et al., 2024; Bamberger et al., 2024; Tang et al., 2024; Chen et al., 2024b; Qiao et al., 2025; Zheng et al., 2024b; Zhao et al., 2024a; Ekbote et al., 2023; Zheng et al., 2024d;c; Fan et al., 2025; Li et al., 2024b;a; Liao et al., 2024a; Yang et al., 2021; Park et al., 2024; Das et al., 2025; Park et al., 2025a; Abate & Bianchi, 2024; Liao et al., 2024b; Deac & Tang, 2023; Yang & Mirzasoleiman, 2023; Zhang et al., 2025b; Yadati, 2025; Hou et al., 2024; Lim et al., 2024; Mustafa & Burkholz, 2024; Tortorella & Micheli, 2023; Eremeev et al., 2025; Xue & Wu, 2025; Finder et al., 2025; Zheng et al., 2025; Liu et al., 2025b; Luan et al., 2024b; Alkhoury et al., 2025; Wu et al., 2024; Li et al., 2025g; Coşkun et al., 2024; Ceni et al., 2025; Lin et al., 2024; Mendelman et al., 2025; Eliasof et al., 2025; Zheng et al., 2024a; Hevapathige et al., 2025; Luo et al., 2025a; Karabulut & Baytaş, 2024; Ribeiro et al., 2025; Zhou et al., 2024a; Linkerhägner et al., 2024; Hoffmann et al., 2025; Pirrò, 2023; Anson et al., 2024b; Zhang et al., 2025a; Li et al., 2025d; Maekawa et al., 2023; Liu et al., 2025a; Ai et al., 2025b; Li et al., 2025a; Fiorini et al., 2025; Gupta et al., 2025; Kohn et al., 2024; Xu et al., 2024b; Chen et al., 2024c; Francesco et al., 2024; Yang et al., 2023; Achten et al., 2024; Aliakbari et al., 2024; Anonymous, 2025; Anson et al., 2024a; Islam et al., 2025; Luo et al., 2022; Yu & Gao, 2022; Wei et al., 2023; Qin et al., 2022; Tola et al., 2024; Wang & Fan, 2024; Eliasof et al.; Liu et al., 2024a; Li et al., 2025e; Yu et al., 2024; Wang & Fan, 2024; Sun et al., 2024b; Luo et al., 2025c; Yao & Li, 2024; Velingker et al., 2023; Luo et al.,

2025b; Wei et al., 2023; Yin et al., 2024; Li et al., 2025f; Wei et al., 2021; Sanchez-Martin et al., 2024).

Additionally, we would like to point out an interesting observation. In research works involving anomaly detection on the imbalanced Questions dataset, in addition to regular AUROC, several works report additional metrics such as Average Precision and AUPRC, highlighting the importance of studying different metrics while studying performance of a model (Zhao et al., 2025; Ye et al.; Wang et al., 2025; Peng et al., 2025b; Ai et al., 2025a; Wu & Gu, 2025; Liu et al., 2025c; Qiao et al., 2025; Wei et al.; Tang et al., 2023; Liu et al., 2024b; Do & Ta, 2025; Cheng et al., 2024; PESCE, 2023; Liu et al., 2023; Peng et al., 2025a; Li et al., 2025b).

## B  METRICS

### B.1  AGGREGATE METRICS

**Accuracy (ACC)**  Accuracy, defined as $\text{ACC} = \frac{\text{TP+TN}}{\text{TP+TN+FP+FN}}$, represents the proportion of correctly classified instances out of the total. Here TP, TN, FP, FN refer to True Positive, True Negative, False Positive and False Negative respectively. It is commonly used in several GNN works, even when datasets are imbalanced. It can be misleading in scenarios with significant class imbalance, as a high score can be achieved by simply classifying the majority class correctly.

**Balanced Accuracy**  Balanced Accuracy is defined as the average of recall obtained on each class (Balanced Accuracy $= \frac{1}{N_{\text{classes}}} \sum_{i=1}^{N_{\text{classes}}} \text{Recall}_i$). Balanced Accuracy is crucial when dealing with imbalanced datasets and no specific class is prioritized, as it prevents misleadingly high accuracy scores achieved by models that only perform well on majority classes. By averaging the recall of each class, it robustly reflects a model's true effectiveness in identifying instances across all categories, ensuring fair performance evaluation and generalizability even when class distributions are uneven.

**Area Under the Receiver Operating Characteristic Curve (AUROC)**  AUROC is a threshold-independent metric that quantifies the diagnostic ability of a binary classifier by plotting the True Positive Rate (Recall) against the False Positive Rate (FPR) across all possible thresholds. This metric provides a robust measure of a classifier's ability to distinguish between classes irrespective of a specific decision boundary and offers insights into the model's ranking capabilities.

**Area Under the Precision-Recall Curve (AUPRC)**  AUPRC is a threshold-independent metric that plots Precision (positive predictive value) against Recall (sensitivity) across all possible classification thresholds. It is particularly informative and reliable for tasks where the positive class (often the minority or the class of interest) is rare. AUC-PR provides a more accurate reflection of performance in such cases, as both precision and recall focus on the positive class, making it less susceptible to inflation by abundant true negatives. A higher AUPRC indicates a better ability to retrieve relevant instances without many false positives McDermott et al..

**F1-Score**  The F1-score, calculated as $\text{F1} = 2 \times \frac{\text{Precision} \times \text{Recall}}{\text{Precision} + \text{Recall}}$, is the harmonic mean of Precision and Recall, providing a single metric that balances both.

### B.1.1  PER-CLASS METRICS

To gain fine-grained insights into model behavior and identify specific strengths or weaknesses for individual classes — particularly for minority classes or those with particular significance — we also report per-class metrics derived from the confusion matrix.

**Precision (Per-Class)**  For each class $i$, precision ($\text{Precision}_i = \frac{\text{TP}_i}{\text{TP}_i + \text{FP}_i}$) represents the ratio of correctly predicted positive observations to the total predicted positive observations for that class. This metric is essential for understanding the false positive rate for a specific class; a high precision for a class indicates that when the model predicts that class, it is usually correct.

**Recall (Per-Class)**   For each class $i$, recall ($\text{Recall}_i = \frac{\text{TP}_i}{\text{TP}_i + \text{FN}_i}$) is the ratio of correctly predicted positive observations to all actual observations belonging to that class. Recall is crucial for evaluating the false negative rate for each class. A high recall indicates that the model is able to find most of the actual instances of that class.

**F1-Score (Per-Class)**   The F1-score for each individual class provides a single, balanced metric by computing the harmonic mean of its precision and recall. This offers a more complete picture of performance for that specific class than precision or recall alone.

## C  PARAMETERS

**Node classification:**   For MPNNs, namely GCN, GAT and GraphSAGE, we set hidden layer to 128, dropout to 0.2, varied $\#layers \in \{1, 3, 4, 5, 7, 10\}$ (for ogbn-arxiv $\#layers \in \{1, 3, 5, 7\}$), #epochs to 2000, learning rate to 0.001, with batchnorm and residual connections. For FSGNN, we varied $\#layers \in \{1, 3, 5\}$ and feature-type $\in \{all-features, homophily, heterophily\}$. For Nodeformer and SGFormer, we set heads=2 and varied $\#layers \in \{1, 5\}$.

**Graph classification:**   For MPNNs, namely GCN, GAT and GraphSAGE, we set hidden layer to 128, dropout to 0.2, varied $\#layers \in \{1, 3, 4, 5, 7\}$, pooling $\in \{mean, add\}$ #epochs to 100 for ogbg-molhiv and 300 for COLLAB and COX2, and learning rate to 0.001.

For Graph Transformer GraphGPS and Subgraphormer, we set hidden dimension=64, $pooling \in \{mean, add\}$, dropout=0.2, heads=4, epochs=100. For GraphGPS, we varied $\#layers \in \{1, 5, 10\}$ for ogbg-molhiv, $\{1, 5\}$ for COX2 and COLLAB. For Subgraphormer, we used hidden dimension=128 and layers=5 for ogbg-molhiv and COX2. On COLLAB, we got Out of Memory error for Subgraphormer hence could not report results on it.

For each model, we choose the best configuration based upon validation metrics. By default, following existing literature, we choose these metrics as accuracy for Amazon-ratings, Squirrel-filtered, COLLAB and ogbn-arxiv. We choose AUROC as validation metric for Questions, COX2, and ogbg-molhiv as per the literature. We study in Fig. 4 how changing this metric has an impact on performance.

# D CLASS LEVEL STATISTICS OF OGBN-ARXIV

Due to space limitations in the main paper, we present here the class level statistics of ogbn-arxiv in Table 3.

| Class | Full Dataset | Test Split |
|---|---|---|
| $C_0$ | 0.0033 | 0.0011 |
| $C_1$ | 0.0041 | 0.0038 |
| $C_2$ | 0.0286 | 0.0151 |
| $C_3$ | 0.0123 | 0.0135 |
| $C_4$ | 0.0346 | 0.0385 |
| $C_5$ | 0.0293 | 0.0256 |
| $C_6$ | 0.0096 | 0.0128 |
| $C_7$ | 0.0035 | 0.0028 |
| $C_8$ | 0.0368 | 0.0257 |
| $C_9$ | 0.0167 | 0.0071 |
| $C_{10}$ | 0.0465 | 0.0299 |
| $C_{11}$ | 0.0044 | 0.0049 |
| $C_{12}$ | 0.0002 | 0.0001 |
| $C_{13}$ | 0.0139 | 0.0129 |
| $C_{14}$ | 0.0035 | 0.0015 |
| $C_{15}$ | 0.0024 | 0.0018 |
| $C_{16}$ | 0.1613 | 0.2156 |
| $C_{17}$ | 0.0030 | 0.0042 |
| $C_{18}$ | 0.0044 | 0.0043 |
| $C_{19}$ | 0.0170 | 0.0086 |
| $C_{20}$ | 0.0123 | 0.0064 |
| $C_{21}$ | 0.0023 | 0.0010 |
| $C_{22}$ | 0.0112 | 0.0079 |
| $C_{23}$ | 0.0167 | 0.0166 |
| $C_{24}$ | 0.1310 | 0.2210 |
| $C_{25}$ | 0.0074 | 0.0098 |
| $C_{26}$ | 0.0272 | 0.0214 |
| $C_{27}$ | 0.0284 | 0.0425 |
| $C_{28}$ | 0.1264 | 0.0586 |
| $C_{29}$ | 0.0025 | 0.0025 |
| $C_{30}$ | 0.0698 | 0.0953 |
| $C_{31}$ | 0.0167 | 0.0184 |
| $C_{32}$ | 0.0024 | 0.0017 |
| $C_{33}$ | 0.0075 | 0.0045 |
| $C_{34}$ | 0.0465 | 0.0291 |
| $C_{35}$ | 0.0007 | 0.0007 |
| $C_{36}$ | 0.0208 | 0.0129 |
| $C_{37}$ | 0.0140 | 0.0099 |
| $C_{38}$ | 0.0089 | 0.0044 |
| $C_{39}$ | 0.0120 | 0.0055 |

Table 3: Class wise statistics of ogbn-arxiv.

# E NUMERICAL VALUES FOR DIVERGENCE STUDY OF FIGURE 3

| Model | C1 | C2 | AUROC (C1) | AUROC (C2) | AUPRC (C1) | AUPRC (C2) |
|---|---|---|---|---|---|---|
| GCN | L=3 | L=10 | $0.7581 \pm 0.0125$ | $0.7683 \pm 0.0083$ | $0.2160 \pm 0.0183$ | $0.1924 \pm 0.0196$ |
| SAGE | L=3 | L=10 | $0.7143 \pm 0.0162$ | $0.7510 \pm 0.0155$ | $0.1941 \pm 0.0207$ | $0.1957 \pm 0.0247$ |
| GAT | L=5 | L=10 | $0.7463 \pm 0.0141$ | $0.7467 \pm 0.0130$ | $0.1965 \pm 0.0296$ | $0.1800 \pm 0.0182$ |

Questions

| Model | C1 | C2 | AUROC (C1) | AUROC (C2) | AUPRC (C1) | AUPRC (C2) |
|---|---|---|---|---|---|---|
| GCN | L=4(A) | L=3(A) | $0.7480 \pm 0.0106$ | $0.7444 \pm 0.0276$ | $0.2281 \pm 0.0224$ | $0.2827 \pm 0.0140$ |
| SAGE | L=5(A) | L=4(A) | $0.7610 \pm 0.0116$ | $0.7604 \pm 0.0202$ | $0.2931 \pm 0.0237$ | $0.2926 \pm 0.0154$ |
| GAT | L=1(M) | L=3(A) | $0.7196 \pm 0.0058$ | $0.7192 \pm 0.0175$ | $0.1849 \pm 0.0083$ | $0.2653 \pm 0.0101$ |

ogbg-molhiv

| Model | C1 | C2 | Accuracy (C1) | Accuracy (C2) | Bal. Acc (C1) | Bal. Acc (C2) |
|---|---|---|---|---|---|---|
| GCN | L=3 | L=5 | $0.7192 \pm 0.0013$ | $0.7254 \pm 0.0019$ | $0.4967 \pm 0.0057$ | $0.5191 \pm 0.0037$ |
| SAGE | L=5 | L=7 | $0.7236 \pm 0.0024$ | $0.7228 \pm 0.0024$ | $0.5066 \pm 0.0064$ | $0.5129 \pm 0.0075$ |
| GAT | L=5 | L=7 | $0.7206 \pm 0.0019$ | $0.7200 \pm 0.0027$ | $0.4976 \pm 0.0060$ | $0.5008 \pm 0.0026$ |

ogbn-arxiv

| Model | C1 | C2 | Accuracy (C1) | Accuracy (C2) | Bal. Acc (C1) | Bal. Acc (C2) |
|---|---|---|---|---|---|---|
| SAGE | L=7(A) | L=3(M) | $0.8578 \pm 0.0266$ | $0.8598 \pm 0.0077$ | $0.6567 \pm 0.0424$ | $0.6706 \pm 0.0091$ |
| GAT | L=7(M) | L=4(A) | $0.8598 \pm 0.0091$ | $0.8599 \pm 0.0137$ | $0.6210 \pm 0.0545$ | $0.6680 \pm 0.0134$ |
| GCN | L=4(M) | L=7(A) | $0.8745 \pm 0.0132$ | $0.8706 \pm 0.0120$ | $0.6580 \pm 0.0441$ | $0.7024 \pm 0.0470$ |

COLLAB

(a) Numerical values for divergence study of Figure 3. (A) stands for add pool and (M) stands for mean pool. L refers to number of layers. C1 stands for the first configuration and C2 for the second.

# F LLM USAGE

We disclose that we used LLMs for improving the writing of the paper, specifically for rephrasing sentences.

