# OpenReview forum: "Unpacking Evaluation Pitfalls on Standard GNN Benchmarks"
_ICLR.cc/2026/Conference — Submitted to ICLR 2026_

### Official Review · Reviewer_xAoy · 2025-10-15

**Soundness:** 2
**Presentation:** 3
**Contribution:** 2
**Rating:** 2
**Confidence:** 4

**Summary:**

This paper argues that existing benchmarks for node and graph classification have focused on metrics, such as accuracy, that do not capture the class imbalance in the datasets considered. They argue in favor of the use of metrics, such as balanced accuracy and AUPRC, that better capture the behavior of GNNs on samples of the minority class. Empirically, the authors show that GNNs trained with standard protocols favor examples of the majority classes, as one would intuitively expect.

As the authors themselves put it, “This gap is not merely a statistical nuance; it profoundly limits our understanding of a model’s true generalization capabilities and its reliability in real-world, high-stakes applications like drug discovery or anomaly detection”.

**Strengths:**

The main strength of the paper is giving a voice to the often-neglected aspect of proper empirical evaluations in the GNN space, with emphasis on the class imbalance aspect. The article is mostly well written and employs a clear organization that makes it easy to understand and follow.

The paper feels like a very linear position paper, as it describes the problem of improper use of evaluation metrics for a selected number of common datasets, provides empirical evidence that current GNNs have not been thoroughly evaluated with respect to the class imbalance of datasets. I would not say that this paper proposes to perform a careful re-evaluation of GNNs considering the imbalance problem, as the training procedure itself has not been set up to take into account imbalance (but for the early stopping analysis at the end of Section 4). As such, the following comments will reflect this view that the paper wants to point out the problem rather than address it.

**Weaknesses:**

Overall, I think this is a good paper, so the decision boils down to one question: “how impactful is this paper for the GNN community?” This question is particularly pertinent because the paper makes an excellent observation, but the results do not add much more to that in my opinion. This is the biggest limitation. I find that the text re-iterates many times on the concept that it is important to use the right metrics; however, it is very much expected to see such a big gap between accuracy and class imbalance-aware metric when an ML model is trained irrespective of such imbalance.

In this sense, it is my personal view that the paper should provide a more substantial contribution if we still want to believe ICLR is a top-conference and reviews try to improve submissions. An observation alone cannot be enough. More concretely, I believe that the paper could possibly (just suggestions, not requests for the rebuttal):
- Provide precise evaluation guidelines for future works to help researchers correctly validate their models. These pertain to data splitting (e.g., stratified), how GNNs should be trained (e.g., early stopping based on better metrics, but also imbalance-aware training strategies to assess the true ability of models of capturing the class imbalance), etc.
- With these evaluation guidelines in mind, perform a careful empirical re-evaluation of a few base models as well as SOTA ones on these datasets, including a more fine-grained hyper-parameter selection that considers different sizes for the hidden layer, for instance.

Discussing the first point only would already greatly improve the usefulness of the paper, but it still does not bring it to the expected level for an A*-level conference. Without a strong re-evaluation that addresses the highlighted problem, I would suggest that the authors submit to a less prestigious venue.

**Questions:**

Questions:
- Is the standard 80/10/10 split of previous works stratified? If not, I would expect this could significantly bias the results shown in the paper.
- Why did the authors not perform hyperparameter tuning for the hidden layer size?
- What is the added value of Figure 3 compared to Figure 1?

Some further comments:
- I find it interesting that some models like NodeFormer display a different accuracy on misrepresented classes, but it would be nice to see these results on a more consolidated framework. The results may suggest that NodeFormer, for instance, is not capturing well the class imbalance, but a proper training that accounts for it might completely change the ranking. This is one more reason why I believe that the empirical results in the paper do not represent an as impactful contribution as the main problem the authors pointed out.
- Appendix B can be substantially improved, starting with the definition of Precision and Recall and trying to keep a uniform terminology across the different metrics. For instance, “Recall” is first called True Positive Rate and then sensitivity, which does not help the reader fix the concept in mind.
- Please try to give proper credit, in the introduction and related work section, to the inventors of GNNs, namely the GNN of Scarselli et al. (2009, TNNLS) and the NN4G of Micheli (2009, TNNLS)

---

> ### Author Response · Authors · 2025-12-03
>
> We sincerely thank the reviewer for recognizing the importance of our 'excellent observation' and finding this paper good. We are also thankful for the very detailed feedback. We address the points raised by the reviewer below.
>
>
>
>
> **Parameter and Loss function tuning: Hyperparameters studied for experiments in the rebuttal**
>
> In the rebuttal, we expand our hyperparameters as below. Typically, more hidden layers tuning, and also tuning with respect to loss function(imbalance and ordinal aware), apart from existing studied parameters.
>
> *MolHIV:* We use a learning rate of 0.0001 and dropout of 0.2. Readout uses mean or add pooling. For GCN, we completed with layers {1, 3, 4, 5, 7} with hidden dimensions {64, 128, 256, 512}. For GAT and SAGE, we completed with layers {1, 3, 5, 7, 10} and hidden dimensions {128, 256}, with num_heads=4 for GAT. All MolHIV models are tuned with (standard, focal loss), and validation is done using ROC-AUC and PR-AUC.
>
> *Questions:* For GCN, GAT, and SAGE, we use a learning rate of 0.001, dropout of 0.2, layers {1, 3, 5, 7, 10}, and hidden dimensions {64, 128, 256}. Loss functions (standard, focal) and validation metrics (ROC-AUC, PR-AUC) remain consistent across architectures.
>
> *Amazon & Squirrel-filtered:* Both datasets test layers {1, 3, 5, 7, 10}, dimensions {64,128,256,512}, and dropout {0.5, 0.7} using Cross Entropy (CE) and Distance Weighted Cross Entropy (DWCE, alpha 0.1, 0.5) losses for Acc/QWK metrics. Amazon-ratings uses a learning rate of 0.001, while Squirrel-filtered uses 0.01. Validation metrics: Accuracy and QWK(ordinal nature of the dataset labels). FSGCN, uses all hyperparameters from the Amazon and Squirrel-filtered configurations above, with the addition of setting feature-types=all.
>
>
>
> ### **Proposed Evaluation guidelines:**
> We thank the reviewer for this comment.
>
> Based upon the evaluation results shown in response to *bwF3*(AUCPR vs AUROC -> Top-k) on ogbg-molhiv and questions, ordinality results(Amazon and Squirrel-filtered) to reviewer *vz42*, and rank inversion results to reviewer *TdYK*, we present the following evaluation guidelines.
>
> **1. Utility metrics for High-Stakes Retrieval Tasks (e.g., ogbg-molhiv):** For tasks where false positives are costly (e.g., drug discovery,  ogbg-molhiv), we propose that future works also prioritize utility metrics such as  (NDCG@K, PREC@K) over just AUROC. In scenarios such as drug discovery false positives are costly. Our evaluation shows that while a GCN model selected via the standard ROCAUC achieved the highest score on AUROC(0.7844), it performed poorly on downstream top-k metric (NDCG@1%: 0.3232). Conversely, a SAGE model selected via our proposed PRAUC metric—despite a lower ROC score—yielded obtained a higher utility in the top 1% (NDCG@1%: 0.6011). Similar results hold for different K as well and also on PREC@K metric. This confirms that on datasets such as ogbg-molhiv, additional metrics as *top-k* should be also studied. Further, from the current experiments, stopping criteria using AUCPR can lead to better top-k retrieval.
>
>
> **2. Ordinality:** Second, for ordinal classification datasets like Amazon-Ratings and Squirrel-filtered(ordinal labels 1-5), we advocate for the inclusion of ordinal-aware metrics alongside standard Accuracy. Accuracy treats all errors equivalently, penalizing "near misses" (e.g., predicting 4 stars for a 5-star item) the same as "catastrophic misses" (e.g., predicting 1 star). Our experiments reveal that high accuracy does not guarantee that the model has learned the underlying class order. Reporting QWK alongside accuracy provides a more holistic evaluation aligned with the ordinal nature of these datasets.
>
> **Cost of Mistakes:** Finally, we recommend that future studies quantify the severity of misclassifications by reporting the "Cost of Mistakes" (MAE calculated on failure cases). While accuracy measures hit-rate, it ignores how "wrong" a prediction is. Our analysis shows that optimizing for QWK significantly reduces the magnitude of errors. Hence, we argue that reporting the Cost of Mistakes provides critical insight into model reliability that accuracy alone cannot capture.
>
>
> **3. Training objective :** Finally, we find that these evaluation guidelines must extend to the training objective itself. We recommend utilizing Ordinal-Aware Loss functions, such as Distance Weighted Cross Entropy (DWCE), rather than standard Cross-Entropy (CE), which treats ordinal labels as independent categories. Our isolation study on Amazon-Ratings showed that switching from Cross-entropy to DWCE(Class distance weighted cross entropy) improved downstream ordinality metrics and reduced the cost of errors.

---

> ### Author Response · Authors · 2025-12-03
>
> **Q. Stratified split:**
> *Response*: For benchmark datasets, we use the standard splits provided and which are followed by the existing GNN community. For others, we use stratified splits.
>
>
> **Q. Value of Figure 3 compared to Figure 1?**
> *Response*: We have now further, discussed this w.r.t impact of optimization objective on downstream task, specifically w.r.t AUCPR vs AUROC. As discussed in response to reviewer bwF3, in certain applications such as drug discovery  in datasets such as ogbg-molhiv, utilizing AUPRC as the stopping criterion captures a model with better retrieval capabilities(Top-K).
>
>
> **Q. Recall vs TPR in Appendix**
> We thank the reviewer for this comment, and we will update to make it consistent.
>
>
> **Q. Citation:**
> We thank the reviewer for pointing this. We will add the below citations in the introduction.
>
> [1] Scarselli, Franco, et al. "The graph neural network model." IEEE transactions on neural networks 20.1 (2008): 61-80.
>
> [2] Micheli, Alessio. "Neural network for graphs: A contextual constructive approach." IEEE Transactions on Neural Networks 20.3 (2009): 498-511.
>
>
> **Conclusion:** We are grateful to the reviewer for their constructive feedback, which pushed us to move beyond the initial observation and conduct a deeper study. We believe the analysis provided above—specifically the rigorous hyperparameter tuning, the formulation of task-specific evaluation guidelines, and the quantification of error costs—has significantly improved the paper.

---

### Official Review · Reviewer_TdYK · 2025-10-15

**Soundness:** 3
**Presentation:** 3
**Contribution:** 3
**Rating:** 6
**Confidence:** 5

**Summary:**

The paper argues that metric typically reported in graph machine learning research on standard graph datasets are poorly fit for the job. Specifically, as many of these standard datasets exhibit significant class imbalance, typically reported metrics such as accuracy for multiclass classification and AUROC for binary classification are unreliable, while metrics such as balanced accuracy and AUPRC provide more meaningful results. The paper shows that when models are evaluated with balanced accuracy and AUPRC, model performance appears to be much further from perfect than standard metrics suggest. Further, using balanced accuracy and AUPRC can uncover strong performance differences between models that appear to have nearly identical performance when evaluated with standard metrics. The paper also advocates the use of per-class metrics to obtain fine-grained insights into model behavior.

**Strengths:**

The paper addresses an important issue of selecting proper metrics for performance evaluation and can be viewed as part of a recent line of works criticizing the evaluation practices in graph machine learning (e.g., [1-3]). While the main message of the paper is very simple (when a dataset exhibits strong class imbalance, researchers should use appropriate evaluation metrics), it is an important message, and it is worth to bring it to the attention of the graph machine learning community. The paper highlights how unreliable modern evaluation practices can be. The experiments in the second part of Section 4.2 (Divergent Metrics paragraph) are interesting: they show that while two model modifications might appear performing almost identically under accuracy or AUPRC, using metric that are more appropriate for class-imbalanced datasets can show that there is in fact substantial difference between these models.



[1] Graph Learning Will Lose Relevance Due To Poor Benchmarks (ICML 2025)

[2] No Metric to Rule Them All: Toward Principled Evaluations of Graph-Learning Datasets (ICML 2025)

[3] GraphLand: Evaluating Graph Machine Learning Models on Diverse Industrial Data (NeurIPS 2025)

**Weaknesses:**

- The paper does not discuss why metrics taking class imbalance into account can be more appropriate for model evaluation in practical applications. For example, in the setting of multiclass classification, if an incorrect prediction for any object leads to equal negative outcomes (e.g., monetary losses), then accuracy is a perfectly valid metric. However, if errors for objects from smaller classes lead to larger negative outcomes (e.g., from the point of view of business objectives or societal fairness), then balanced accuracy would be a better metric. I believe discussing these issues with concrete examples from popular datasets will significantly strengthen the paper.

- The range of the datasets considered in the paper is rather narrow. While almost all popular datasets in graph machine learning exhibit class imbalance, the paper only considers a few of them. For example, only 4 datasets are considered for multiclass classification, and I believe one of them (squirrel-filtered) should not actually be used. This dataset was introduced in [4] to show that the original squirrel dataset is flawed because it has duplicate nodes and removing these nodes (thus obtaining squirrel-filtered) significantly changes model performance and ranking. However, it is not known what exactly is the source of duplicate nodes in the original dataset and whether removing them is meaningful. Thus, instead of using squirrel-filtered, [4] proposes to use an entirely new set of datasets for model evaluation under heterophily (and some of them are used in this work). Thus, I suggest not considering squirrel-filtered and replacing it with more meaningful datasets. For binary classification, only 3 datasets are considered. I suggest looking at more datasets from the MoleculeNet suit of molecular graphs (these datasets are also available via OGB). For example, the MUV dataset (mol-muv in OGB) has 17 binary prediction tasks each with strong class imbalance, while almost all works use AUROC to evaluate model performance on it, which is not an appropriate metric.

- While the experiments in the second part of section 4.2 are interesting (as discussed above), the experiments in the first and third parts are not so interesting. The third part (Optimizing for different objectives) simply shows that if we optimize a certain metric A, then we will get better values of metric A than if we optimize some other metric B, which is a very expected result. The first part (Performance variation when taking class imbalance into consideration) shows that the absolute values of balanced accuracy or AUROC are lower than the absolute values of accuracy or AUPRC, respectively. However, these values are not directly comparable (since the metrics are different and have different interpretations). It would be more interesting to investigate whether model rankings change under different metrics (it is partly shown in the second part, but only for 2 variations of the same model, not for more variations or between different models). Overall, I believe providing more experiments that show how using inappropriate metrics can lead to making suboptimal decisions (e.g., choosing a model that is not the best) will significantly strengthen the paper.



I am willing to raise my score if my concerns are addressed.



[4] A critical look at the evaluation of GNNs under heterophily: Are we really making progress? (ICLR 2023)

**Questions:**

See weaknesses.

Minor details:

- Lines 319-323 repeat the same thing twice.

- "On several imbalanced ogbn-arxiv node classification datasets (Fig. 1e)" - there seem to be erroneously omitted and/or inserted words here.

- While the whole paper is dedicated to proper metrics for datasets with imbalanced classes, this is not reflected in the paper title. I suggest making the paper title more specific.

---

> ### Author Response · Authors · 2025-12-03
>
> We sincerely thank the reviewer for their positive assessment and for recognizing that our work addresses an "important message" regarding the reliability of modern evaluation practices.  We are also thankful for the very detailed feedback. We also appreciate the valuable references placing our work within the recent line of literature scrutinizing GNN evaluation standards. We have addressed your specific concerns below:
>
> ### **Linking metrics to practical applications**
>
> We thank the reviewer for this insightful observation. As the reviewer correctly notes, when errors on minority classes or specific outcomes carry disproportionate "negative outcomes" (whether monetary or societal), the evaluation metric must reflect this asymmetry. To strengthen the paper as suggested, we have expanded our analysis to explicitly map the **"Cost of Error"** profiles for our datasets, demonstrating exactly why standard metrics are misaligned with deployment objectives:
>
> ### **1. Retrieval & High-Stakes Tasks**
> * **Dataset:** `ogbg-molhiv` (Molecular Property Prediction)
> * **Practical Setting:** This task mirrors **Virtual Screening** in pharmaceutical pipelines. While large number of compounds are ranked computationally, downstream physical experimentation ("wet-lab" validation) is restricted to a fixed, small budget (e.g., only the top-100 candidates are tested).
> * **Asymmetric Outcome:**
>     * **False Positive in Top-K (High Cost):** Ranking an inactive molecule highly leads to the misallocation of limited laboratory resources and significant monetary and time loss.
> * **Analysis:** : **AUPRC** takes the context into consideration (and ranking metrics like NDCG@K and PREC@K) specifically penalize the model for failing to enrich the top-ranked candidates[1].
>
> Note: Further, we have also discussed the implications of AUPRC and AUROC in response to reviewer bwF3, on how the quality of top-k predictions changes when using AUPRC and AUROC as stopping criteria. We observe that, models chosen by AUPRC over AUROC, obtain better performance on *top-k(ndcg@k, precision@k)* retrieval of minority classes on average in molhiv(as well as questions).
>
> ### **2. Severity of Error: Ordinal Classification**
> * **Datasets:** `Amazon-Ratings`, `Squirrel-filtered`
> * **Practical Setting:** Predicting ordinal values, such as user satisfaction ratings (1–5 stars) or discrete traffic density levels.
> * **Asymmetric Outcome:**
>     * **High-Severity Error:** Predicting 1-star for a 5-star product. This could represent a failure to capture the ordinal signal.
>     * **Low-Severity Error:** Predicting 4-stars for a 5-star product. This error magnitude is small in comparison to the high-severity error.
>
> * **Analysis:** Standard **Accuracy** employs a 0/1 loss function, treating a "High-Severity" error (Distance=4) identically to a "Low-Severity" error (Distance=1). To address this, we advocate for metric such as **Quadratic Weighted Kappa (QWK)**. By penalizing errors based on their magnitude, QWK forces the model to minimize the *severity* of its mistakes. By evaluating with such ordinal metrics, we gain insight into the model's reliability and the "Cost of Mistakes," in better understanding how far from the truth the prediction is.
>
> Note: In our response to reviewer vz42, we observe that models optimized for accuracy not always perform best in maintaining the ordinal relationship. Hence studying, ordinal metrics such as QWK during evaluation, help in better understanding of the utility of the model along with cost of errors as cost of wrong prediction from 5->1 is more severe than 5->4 .
>
> ### **3. Fairness & Representation: Long-Tail Classification**
> * **Dataset:** `ogbn-arxiv`, `COLLAB` (Citation Networks)
> * **Practical Setting:** Classifying scientific literature to organize repositories.
> * **Context:** The dataset exhibits significant skew, with four dominant fields  comprising nearly half the data, while niche fields form a "long tail."
> * **Analysis:** While standard **Accuracy** correctly measures the global hit rate, it is heavily biased toward majority classes. In scenarios where diverse representation is a priority, relying solely on Accuracy can obscure poor performance on less frequent topics. Therefore, reporting metrics such as  **Balanced Accuracy** offers a complementary view, ensuring the model maintains utility across the entire spectrum of subjects—not just the dominant categories.
>
> References:
>
> [1] McDermott, M. B. A., Zhang, H., Hansen, L. H., Angelotti, G., & Gallifant, J. (2024). A closer look at AUROC and AUPRC under class imbalance. NeurIPS, Vol. 37.

---

> > ### Author Response · Authors · 2025-12-03
> >
> > ### **Q. Squirrel-filtered dataset and molmuv**
> >
> > We agree with the reviewer that even the fixed Squirrel-filtered dataset by Plantov et al. might not be ideal. However, because this version remains a standard benchmark in current GNN literature, removing it entirely would hinder direct comparison. We will therefore would include a clear disclaimer in the main text referencing the known issues with this benchmark, ensuring that readers interpret the results with the necessary context while prioritizing the findings from robust datasets like Amazon-Ratings.
> >
> >
> > ### **Q. Regarding expansion to ogbg-molmuv:**
> > We strongly agree with the suggestion to incorporate ogbg-molmuv. As the reviewer notes, this dataset contains 17 distinct tasks with extreme class imbalance, representing a "needle in a haystack" scenario. This makes it an ideal case study for our argument apart from molhiv.
> >
> > **Computational Constraints:**
> > However, we wish to be transparent regarding the feasibility of completing this specific experiment during the rebuttal window while also running other experiments on different datasets. The ogbg-molmuv dataset is computationally demanding; properly training and tuning requires approximately 6-7 hours per parameter configuration on our infrastructure. Given the parameter sweep required to ensure a fair comparison between metrics, we were unable to complete this analysis.
> >
> > **Hypothesis and Commitment:**
> > Despite this constraint, we are actively running these experiments and commit to including ogbg-molmuv in the final version of the paper. We believe the results will strongly reinforce our central thesis. Our current deep-dive into ogbg-molhiv (which shares the same domain and imbalance characteristics) already shows that relying on AUPRC as validation metric on an average leads to better performance on *top-k* retrieval than relying on AUROC. Since molmuv exhibits even more severe imbalance than molhiv, we hypothesize that the discrepancy between AUROC (which rewards easy negatives) and AUPRC (which rewards top-tier hits) will be even more pronounced, further validating the necessity of studying different metrics. However, we would like to let our experiments complete before we making a final conclusion on this dataset.

---

> ### Author Response · Authors · 2025-12-03
>
> ### **Showing Suboptimal Decisions and possible rank inversions**
>
> We thank the reviewer for this critical insight. We fully agree with the assessment that simply showing "optimizing for Metric A improves Metric A" is an expected result. The far more significant research question—which the reviewer correctly identified—is whether relying on inappropriate metrics leads practitioners to make suboptimal decisions, such as selecting a model that performs poorly in actual deployment.
>
> To address this, we conducted a rigorous "Counterfactual Model Selection" study on ogbg-molhiv. We investigated whether the "best" model checkpoint identified by the standard community metric (AUROC) corresponds to the "best" model for high-stakes retrieval (Top-K Utility), or if the rankings invert.
>
> In the context of ogbg-molhiv, the evaluation objective is to select a small shortlist of candidate molecules for expensive experimental validation. In this setting, the cost of a False Positive at the top of the ranking (wasted laboratory resources) significantly outweighs the cost of a False Negative.
>
> **Rank inversion:** As observed in the table below, the GCN model achieving the highest Test AUROC(0.7844)—the community standard—fails to maximize utility in this critical top-tier region(NDCG@1% of 0.3232). In contrast, SAGE model selected via AUPRC achieve significantly higher utility (NDCG@1% of 0.6011 ± 0.05) despite having lower AUROC scores. Consequently, a practitioner relying strictly on AUROC might discard the relevant models, and instead deploy models that are significantly less effective at enriching the candidate list for drug development trials. Crucially, this demonstrates that 'state-of-the-art' performance on a proxy metric (like AUROC) might not always correlate with—performance on another important metric of interest such as NDCG@K or PREC@K.
>
> Hence, we argue that studying *top-k* performance metrics and performance evaluation on different stopping criteria on such datasets help in making a more informed decisions.
>
> *Note:* We have highlighted a similar study to reviewer bwF3, where we observe that models chosen by AUPRC over AUROC, tend to perform better on *top-k* metrics such as NDCG@K and PREC@K on datasets ogbg-molhiv and questions which are heavily imbalanced. This is critical when retrieving *top-k* correctly are important.
>
> **Conclusion:** In summary, our counterfactual analysis on ogbg-molhiv suggest that the models identified as "best" by the community-standard AUROC metric can be suboptimal for high-stakes top-k retrieval. By strictly following standard evaluation protocols, practitioners risk discarding models that offer significantly higher utility in real-world deployment scenarios (e.g., drug screening). These findings reinforce the necessity of the expanded evaluation guidelines proposed in our work.
>
>
> ### **Subset of results to depict rank inversion results on molhiv.**
>
> ### **NDCG@K**
>
> | Architecture | Loss      | Selection Metric      | Test ROC | NDCG@1% | NDCG@2% | NDCG@3% | NDCG@4% | NDCG@5% |
> |--------------|-----------|------------------------|-------------------------|--------------------|---------|---------|---------|---------|
> | SAGE         | Standard  | PRAUC      | **0.7394 ± 0.0227**          | **0.6011 ± 0.05**      | 0.4947 ± 0.04 | 0.4135 ± 0.03 | 0.4178 ± 0.02 | 0.4284 ± 0.03 |
> SAGE | Focal | PRAUC |	**0.7773±0.0035** |	**0.4877±0.0183** |	0.4617±0.0282 |	0.3932±0.0206 |	0.4092±0.0221 |	0.4320±0.0115 |
> | GCN          | Standard  | ROCAUC  | **0.7844 ± 0.0075**          | **0.3232 ± 0.07**      | 0.3721 ± 0.03 | 0.3299 ± 0.03 | 0.3425 ± 0.03 | 0.3791 ± 0.03 |
>
> ### **Prec@K**
>
> | Model | Loss     | Selection Metric | Test ROC | Prec@1%               | Prec@2%               | Prec@3%               | Prec@4%               | Prec@5%              |
> |-------|----------|-----------------|-----------------------|------------------|------------------|------------------|------------------|------------------|
> | SAGE  | Standard | PRAUC           | **0.7394 ± 0.0227**       |**0.5873±0.0449**	|0.4538±0.0316 |	0.3602±0.0201|	0.2909±0.0131	| 0.2427±0.0137 |
> SAGE |Focal 	| PRAUC | 	**0.7773±0.0035** | 	**0.5476±0.0194** | 	0.4819±0.0295 | 	0.3844±0.0212 | 	0.3212±0.0178 | 	0.2783±0.0046 |
> | GCN   | Standard | ROCAUC          | **0.7844 ± 0.0075**       | **0.3889±0.0683** | 	0.4177±0.0227 |	0.3468±0.0263 |	0.2869±0.0200 |	0.2638±0.0150
>  |
>
>
> Detailed result for different models are present in the next response.

---

> ### Author Response · Authors · 2025-12-03
>
> Below we present all the results obtained for each model on molhiv.
>
> ### **GraphSAGE**
>
> **NDCG@K**
> | Loss     | Selection Criteria | Test ROC         | @1%               | @2%               | @3%               | @4%               | @5%               |
> |----------|-----------|------------------|--------------------|--------------------|--------------------|--------------------|--------------------|
> | Focal    | PRAUC     | 0.7773±0.0035    | 0.4877±0.0183      | 0.4617±0.0282      | 0.3932±0.0206      | 0.4092±0.0221      | 0.4320±0.0115      |
> | Focal    | ROCAUC    | 0.7769±0.0056    | 0.4248±0.0723      | 0.3764±0.0307      | 0.3250±0.0268      | 0.3489±0.0243      | 0.3750±0.0350      |
> | Standard | PRAUC     | 0.7394±0.0227    | 0.6011±0.0584      | 0.4947±0.0434      | 0.4135±0.0305      | 0.4178±0.0288      | 0.4284±0.0324      |
> | Standard | ROCAUC    | 0.7748±0.0080    | 0.3625±0.0392      | 0.3962±0.0115      | 0.3421±0.0029      | 0.3545±0.0103      | 0.3703±0.0127      |
>
> **PREC@K**
> | Loss     |  Selection Criteria | Test ROC         | @1%               | @2%               | @3%               | @4%               | @5%               |
> |----------|-----------|------------------|--------------------|--------------------|--------------------|--------------------|--------------------|
> | Focal    | PRAUC     | 0.7773±0.0035    | 0.5476±0.0194      | 0.4819±0.0295      | 0.3844±0.0212      | 0.3212±0.0178      | 0.2783±0.0046      |
> | Focal    | ROCAUC    | 0.7769±0.0056    | 0.4921±0.0786      | 0.3936±0.0248      | 0.3199±0.0201      | 0.2788±0.0171      | 0.2476±0.0221      |
> | Standard | PRAUC     | 0.7394±0.0227    | 0.5873±0.0449      | 0.4538±0.0316      | 0.3602±0.0201      | 0.2909±0.0131      | 0.2427±0.0137      |
> | Standard | ROCAUC    | 0.7748±0.0080    | 0.4524±0.0514      | 0.4498±0.0057      | 0.3602±0.0101      | 0.2970±0.0086      | 0.2524±0.0069      |
>
>
>
>
> ## **GAT**
>
> **NDCG@K**
> | Loss     |  Selection Criteria | Test ROC         | @1%               | @2%               | @3%               | @4%               | @5%               |
> |----------|-----------|------------------|--------------------|--------------------|--------------------|--------------------|--------------------|
> | Focal    | PRAUC     | 0.7393±0.0088    | 0.4712±0.0646      | 0.3970±0.0443      | 0.3467±0.0200      | 0.3495±0.0209      | 0.3565±0.0268      |
> | Focal    | ROCAUC    | 0.7109±0.0088    | 0.5910±0.0769      | 0.4431±0.0544      | 0.3625±0.0450      | 0.3650±0.0416      | 0.3807±0.0498      |
> | Standard | PRAUC     | 0.7621±0.0073    | 0.3714±0.1445      | 0.3554±0.0900      | 0.3071±0.0513      | 0.3279±0.0401      | 0.3611±0.0425      |
> | Standard | ROCAUC    | 0.7656±0.0029    | 0.2475±0.0593      | 0.2926±0.0078      | 0.2702±0.0153      | 0.2906±0.0185      | 0.3240±0.0262      |
>
>
> **PREC@K**
> | Loss     |  Selection Criteria | Test ROC         | @1%               | @2%               | @3%               | @4%               | @5%               |
> |----------|-----------|------------------|--------------------|--------------------|--------------------|--------------------|--------------------|
> | Focal    | PRAUC     | 0.7393±0.0088    | 0.5159±0.0405      | 0.3936±0.0301      | 0.3280±0.0101      | 0.2626±0.0103      | 0.2168±0.0100      |
> | Focal    | ROCAUC    | 0.7109±0.0088    | 0.6032±0.0297      | 0.4016±0.0227      | 0.3091±0.0231      | 0.2485±0.0178      | 0.2136±0.0221      |
> | Standard | PRAUC     | 0.7621±0.0073    | 0.4048±0.1468      | 0.3655±0.0801      | 0.2984±0.0461      | 0.2586±0.0281      | 0.2379±0.0198      |
> | Standard | ROCAUC    | 0.7656±0.0029    | 0.3095±0.0701      | 0.3373±0.0098      | 0.2930±0.0266      | 0.2525±0.0174      | 0.2330±0.0221      |
>
>
> continued

---

> > ### Author Response · Authors · 2025-12-03
> > **Continued Response**
> >
> > continued
> >
> >
> > ### **GCN**
> >
> > **NDCG@K**
> >
> > | Loss     |  Selection Criteria | Test ROC         | @1%               | @2%               | @3%               | @4%               | @5%               |
> > |----------|-----------|------------------|--------------------|--------------------|--------------------|--------------------|--------------------|
> > | Focal    | PRAUC     | 0.7463±0.0229    | 0.5795±0.0374      | 0.5101±0.0372      | 0.4315±0.0181      | 0.4501±0.0227      | 0.4605±0.0190      |
> > | Focal    | ROCAUC    | 0.7609±0.0097    | 0.4208±0.0962      | 0.3647±0.0447      | 0.3119±0.0289      | 0.3233±0.0202      | 0.3425±0.0105      |
> > | Standard | PRAUC     | 0.7370±0.0079    | 0.5476±0.1006      | 0.4324±0.0471      | 0.3711±0.0318      | 0.3898±0.0261      | 0.4089±0.0237      |
> > | Standard | ROCAUC    | 0.7844±0.0075    | 0.3232±0.0699      | 0.3721±0.0324      | 0.3299±0.0344      | 0.3425±0.0331      | 0.3791±0.0296      |
> >
> >
> >
> > **PREC@K**
> >
> > | Loss     |  Selection Criteria | Test ROC         | @1%               | @2%               | @3%               | @4%               | @5%               |
> > |----------|-----------|------------------|--------------------|--------------------|--------------------|--------------------|--------------------|
> > | Focal    | PRAUC     | 0.7463±0.0229    | 0.5476±0.0389      | 0.4699±0.0548      | 0.3790±0.0237      | 0.3212±0.0262      | 0.2670±0.0182      |
> > | Focal    | ROCAUC    | 0.7609±0.0097    | 0.4762±0.0847      | 0.3735±0.0197      | 0.3011±0.0137      | 0.2505±0.0076      | 0.2184±0.0119      |
> > | Standard | PRAUC     | 0.7370±0.0079    | 0.5238±0.1214      | 0.3815±0.0410      | 0.3172±0.0266      | 0.2727±0.0227      | 0.2362±0.0165      |
> > | Standard | ROCAUC    | 0.7844±0.0075    | 0.3889±0.0683      | 0.4177±0.0227      | 0.3468±0.0263      | 0.2869±0.0200      | 0.2638±0.0150      |
> >
> >
> > We thank the reviewer for the additional experiments which further strengthened our work.

---

### Official Review · Reviewer_vz42 · 2025-10-19

**Soundness:** 3
**Presentation:** 3
**Contribution:** 2
**Rating:** 4
**Confidence:** 4

**Summary:**

This work considers widely used GNN graph classification / node classification data sets. It argues that most papers consider accuracy/AUROC as their primary metric which may not be ideal in the face of severe class imbalance, particularly in the case of multi-class classification. They then conduct a series of experiments showing that different metrics lead to different insights / interpretations of the model performances.

**Strengths:**

The paper addresses a real issue that is underdiscussed, namely the overuse of "overall acccuracy" without much thought into the merits of this choice.

**Weaknesses:**

For a paper that aims to improve the field, by promoting better use of evaluation metrics, there is relatively little discussion of the relative merits of each metric. For instance, when should one perfer balanced F1 scores over overall accuracy (and why)? What are the pros and cons of each? The definitions are in Appendix B, but there should be more discussion in the main paper (beyond definitions).

Several of the classification tasks are actually ordinal regression tasks. For instance in Amazon, a mistake of 1 star vs 4 stars is significantly worse than 3 stars vs 4 stars. Similar with squirrel. These data sets, are of course, commonly treated as "classification data sets" but that is part of the problem that this paper, in its best form (future iterations maybe) should address.

Its generally unclear what the contribution of this paper is other than (correctly) pointing out that ML papers commonly ignore experimental best practices, either for the sake of convenience or lack of statistical training. This is a real issue, but to in order for the paper to help remedy the situation, there should be more discussion of which metrics are appropriate in which settings and why

**Questions:**

For squirrel filtered (and other wikipedia derived data sets), the goal is to predict traffic level which, in principal is a a regression task (that is turned into an ordinal regression task by coarse graining in order to make the problem "easier".) Is there enough data publicly available to reformulate this is a regression task? That seems like this would alleviate the issues with accuracy.

---

> ### Author Response · Authors · 2025-12-03
>
> We appreciate the reviewer's acknowledgement of the significance of this problem and thank them particularly on the insightful observation regarding the ordinal nature of datasets like Amazon and Squirrel, and the suggestion to include a deeper discussion on the comparative merits of specific metrics.
> We are also thankful for the very detailed feedback.
>
> ### **Explicit Discussion on relative merits of each metric.**
>
> We thank the reviewer for this critical observation. We agree that the paper significantly benefits from an explicit discussion regarding the rationale and relative merits of these metrics. Below, we provide a detailed comparative analysis of the metrics used, distinguishing their utility based on specific graph learning challenges: class imbalance, retrieval goals, and ordinal structure.
>
> **Class Imbalance : Accuracy vs. Balanced Accuracy**
> Regarding the choice between Overall Accuracy and Balanced Accuracy, the distinction is strictly driven by the extreme class imbalance. Accuracy calculates the ratio of correct predictions globally. In highly imbalanced datasets such as *ogbn-arxiv(40 classes)*, a critical challenge in this dataset is the severe class imbalance : the top 4 most frequent classes are disproportionately dominant, accounting for approximately 48% of the total samples, while the bottom 10 least frequent classes contain less than 2.4% of the samples combined. This imbalance makes accuracy a potentially unreliable indicator of overall model effectiveness, especially for identifying papers in niche or relatively less-represented categories. Hence, studying a metric such as balanced accuracy complements the accuracy metric.
>
> **Regarding AUROC versus AUPRC:** While AUROC evaluates discriminative power uniformly across the score spectrum, this neutrality is detrimental in settings where the cost of false positives significantly outweighs that of false negatives. As illustrated in drug discovery—specifically when selecting candidate molecules from a fragment library for costly experimental validation—the metric must facilitate a reduction in high-score false positives[1]. Our reliance on AUPRC is therefore strictly motivated by the need to correct these high-score errors, a nuance that AUROC’s uniform error treatment obscures, thereby failing to capture improvements in critical, budget-constrained decisions.
>
>
>
>
>
> **Ordinal Structure: Accuracy vs. Quadratic Weighted Kappa (QWK)**
> For datasets with inherent order (e.g., `Amazon-Ratings`, `Squirrel-filtered` both have ordinal classes ranging from 1-5), standard classification metrics fail to capture the *magnitude* of errors.
> * **Standard Accuracy:** Employs a binary 0/1 loss. It treats a "severe" error (predicting "1-star" for a "5-star" product) identically to a "minor" error (predicting "4-stars").
> * **Quadratic Weighted Kappa (QWK):** We utilize QWK to address this misalignment. QWK measures inter-rater agreement for ordinal scales, corrected for chance. Unlike accuracy, it applies a weight matrix that penalizes disagreements quadratically based on the distance between the prediction ($i$) and the ground truth ($j$).
>
> Formally, QWK is defined as:
> $$QWK = 1 - \frac{\sum_{i,j} w_{i,j} O_{i,j}}{\sum_{i,j} w_{i,j} E_{i,j}}$$
>
> Where $O_{i,j}$ is the observed confusion matrix, $E_{i,j}$ is the expected matrix under chance agreement, and the quadratic weights are given by $w_{i,j} = \frac{(i-j)^2}{(N-1)^2}$ [2].
>
> In the QWK metric, "near-misses" are significantly more acceptable than catastrophic failures.
>
> Reference:
>
> [1] McDermott, M. B. A., Zhang, H., Hansen, L. H., Angelotti, G., & Gallifant, J. (2024). A closer look at AUROC and AUPRC under class imbalance. NeurIPS, Vol. 37.
>
> [2] Cohen, J. (1968). Weighted kappa: Nominal scale agreement provision for scaled disagreement or partial credit. *Psychological Bulletin*, 70(4), 213.

---

> > ### Author Response · Authors · 2025-12-03
> >
> > ### **Metric Misalignment (Accuracy vs. Ordinality) on Amazon-Ratings and Squirrel-filtered ordinal datasets.**
> >
> > We thank the reviewer for this critical observation. We strongly agree that treating datasets with inherent order as "flat" classification tasks(the standard followed by the community on Squirrel-filtered and Amazon-Ratings) creates a misalignment between the metric and the real-world cost of errors.
> >
> > ### **1. Dataset Properties & Cost of Error**
> > In **Amazon-Ratings**, the labels (1–5 stars) are ordinal; the "distance" between predictions matters. As the reviewer notes, predicting 1-star for a 4-star product (distance=3) is a significantly "worse" failure than predicting 3-stars (distance=1), yet standard Accuracy penalizes both equally (0/1 loss). Similarly, **Squirrel-filtered** labels represent discretized traffic density(1-5), which is inherently ordinal.
> >
> > To understand this in more detail, we conducted a comprehensive study comparing models optimized for **Accuracy** (standard practice) vs. models optimized for **Quadratic Weighted Kappa (QWK)** and **Distance Weighted Cross Entropy (DWCE)**.
> >
> >
> >  ### **2. The Necessity of Ordinal-Aware Metrics**
> >
> > Our empirical results demonstrate that relying solely on Accuracy is insufficient for datasets with inherent ordering. We find that optimizing for Accuracy often masks a model's failure to capture the ordinal structure, creating a disconnect between "correct classification" and the "severity of error."
> >
> > We observe the following patterns across **GCN, SAGE, GAT, and FSGCN**:
> >
> > * **The "Accuracy Illusion" (High Accuracy $\neq$ High Ordinality):**
> >     Accuracy might not be the ideal metric when the classes have ordinal relationships. A model can maximize exact hits (Accuracy) but might still fail at learning the ordinal relationship between classes. For instance,
> >     * **On Squirrel-filtered (SAGE):** The Accuracy-optimized model appears competitive ($36.65\%$), yet its QWK is critically low ($0.3224$). This implies the model is "binning" correctly often enough to satisfy the Accuracy metric, but its errors are severe. By optimizing for ordinality (using QWK validation metric  and DWCE loss), Test QWK increases to **$0.4983$** ($+0.17$). This shows that the "lower accuracy" model ($32.42\%$)   learned the traffic density distribution better.
> >
> >     * **On Amazon-Ratings (GCN):** Similarly, the accuracy-optimized model reaches a peak of $54.04\%$ but plateaus at a lower QWK ($0.4471$). When explicitly optimized for ordinality, the model accepts a lower accuracy ($50.57\%$) but achieves a superior ranking capability ($0.4759$).
> >
> > * **Consistency Across Architectures (Spatial & Spectral):**
> >     * **GAT on Amazon-Ratings:** High accuracy ($54.99\%$) does not guarantee high ordinal performance ($0.4589$). Switching to a QWK-aware loss (DWCE) boosts QWK to **$0.4810$**.
> >     * **FSGCN on Squirrel-filtered:** While Accuracy drops slightly ($37.03\% \rightarrow 35.22\%$) when optimizing for ordinality, the QWK jumps significantly ($0.4655 \rightarrow \mathbf{0.5250}$), an improvement of $+0.06$.
> >
> > **Conclusion:**
> > We argue that future studies on Amazon-Ratings and Squirrel-filtered discuss metrics such as QWK(ordinality-aware) alongside Accuracy.
> >
> > **Detailed results present in next response**

---

> > > ### Author Response · Authors · 2025-12-03
> > > **Response Continued**
> > >
> > > **Detailed results**
> > >
> > > *Acronyms*
> > >
> > > ce: Cross-entropy
> > >
> > > dwce: Distance Weighted Cross Entropy Loss
> > >
> > > QWK: Quadratic Weighted Kappa
> > >
> > > ### **Amazon-ratings: Overall Performance Summary**
> > >
> > > | GNN        | Val Metric | Loss | Accuracy           | QWK               |
> > > |------------|------------|------|--------------------|-------------------|
> > > | fsgcn      | acc        | ce   | 46.34±0.42%        | 0.3865±0.0071     |
> > > | fsgcn      | acc        | dwce | 46.05±0.68%        | 0.3868±0.0111     |
> > > | fsgcn      | qwk        | ce   | 45.48±0.68%        | 0.4064±0.0076     |
> > > | fsgcn      | qwk        | dwce | 45.16±0.74%        | 0.4024±0.0076     |
> > > | gcn        | acc        | ce   | 54.04±0.65%        | 0.4471±0.0103     |
> > > | gcn        | acc        | dwce | 54.11±0.49%        | 0.4581±0.0082     |
> > > | gcn        | qwk        | ce   | 50.57±0.87%   | 0.4759±0.0107     |
> > > | gcn        | qwk        | dwce | 50.41±1.08%        | 0.4795±0.0139     |
> > > | sage       | acc        | ce   | 54.94±0.47%    | 0.4573±0.0128     |
> > > | sage       | acc        | dwce | 55.11±0.47%        | 0.4579±0.0139     |
> > > | sage       | qwk        | ce   | 52.81±1.29%        | 0.4760±0.0103     |
> > > | sage       | qwk        | dwce | 53.34±0.59%        | 0.4765±0.0101     |
> > > | gat        | acc        | ce   | 54.99±0.53%        | 0.4589±0.0111     |
> > > | gat        | acc        | dwce | 55.11±0.69%        | 0.4620±0.0136     |
> > > | gat        | qwk        | ce   | 53.57±0.92%        | 0.4821±0.0090     |
> > > | gat        | qwk        | dwce | 53.34±1.29%        | 0.4810±0.0072     |
> > >
> > > ### **Squirrel-Filtered: Overall Performance Summary**
> > >
> > > | GNN                  | Val | Loss | Accuracy           | QWK               |
> > > |----------------------|-----|------|--------------------|-------------------|
> > > | fsgcn                | acc | ce   | 37.03±2.30%        | 0.4655±0.0424     |
> > > | fsgcn                | acc | dwce | 37.40±2.58%        | 0.4728±0.0581     |
> > > | fsgcn                | qwk | ce   | 35.78±1.69%        | 0.5239±0.0226     |
> > > | fsgcn                | qwk | dwce | 35.22±1.77%        | 0.5250±0.0306     |
> > > | gcn                  | acc | ce   | 44.58±2.18%        | 0.6130±0.0318     |
> > > | gcn                  | acc | dwce | 44.29±2.18%        | 0.6121±0.0307     |
> > > | gcn                  | qwk | ce   | 39.28±3.28%        | 0.6347±0.0240     |
> > > | gcn                  | qwk | dwce | 40.58±4.15%        | 0.6350±0.0259     |
> > > | sage                 | acc | ce   | 36.65±2.16%        | 0.3224±0.0475     |
> > > | sage     | acc | dwce | 36.94±1.15%        | 0.2833±0.0323     |
> > > | sage  | qwk | ce   | 32.42±3.83%        | 0.4983±0.0267     |
> > > | sage                 | qwk | dwce | 31.33±3.32%        | 0.4849±0.0313     |
> > > | gat                  | acc | ce   | 39.80±3.05%        | 0.5053±0.0486     |
> > > | gat                  | acc | dwce | 39.44±2.26%        | 0.4999±0.0404     |
> > > | gat                  | qwk | ce   | 37.35±3.45%        | 0.5450±0.0251     |
> > > | gat                  | qwk | dwce | 39.04±1.68%        | 0.5470±0.0378     |

---

> > > > ### Author Response · Authors · 2025-12-03
> > > > **Part 2: Analysis of cost of mistakes**
> > > >
> > > > ### **3. Analysis of "Cost of Mistakes" (MAE on Failures)**
> > > > To directly address the reviewer's point about the "severity" of errors (e.g., 1 vs 4 stars), we analyzed the Mean Absolute Error (MAE) specifically on samples where the model predicted incorrectly. This quantifies the cost of a mistake.
> > > >
> > > > **Analysis of Amazon-ratings**
> > > >
> > > > On the Amazon-ratings dataset, which uses 5 ordinal star ratings, optimizing for Quadratic Weighted Kappa (QWK) significantly reduced the magnitude of errors (MAE) on the most difficult minority classes (Class 3, 8% of data, and Class 4, 4% of data) compared to optimizing for Accuracy. For the minority  Class 4 (4% of the data), the GCN model exhibited the largest raw MAE under standard CE (Acc) at 2.764 steps away from the true rating. By switching to QWK-based selection, GCN achieved a substantial MAE reduction of 19.50% (2.764 to 2.225 MAE, using CE loss), confirming the conclusion that QWK optimization forces the model to make "safer" errors. Similarly, for the second minority class, Class 3, GCN again showed the most dramatic safety gain, with a relative MAE reduction of 12.20% when optimized for CE (QWK) over CE (Acc). While GAT and SAGE also showed considerable MAE reductions for C4 (approx. 10.5–11.1%), the model FSGCN provided the absolute safest classification for Class 3 when incorporating the DWCE loss and QWK metric, achieving the lowest overall MAE (1.472) for C3 among all models, representing a 6.5% reduction relative to its CE (Acc) baseline.
> > > >
> > > > **Analysis of Squirrel-filtered**
> > > >
> > > > The Squirrel-filtered dataset, representing discretized traffic density (Classes 1–5), showed the most drastic performance gap when switching optimization objectives, particularly for models that failed severely under the standard Accuracy metric. The SAGE model, whose low QWK (0.3224) previously indicated highly severe errors when optimized for Accuracy, showed the most substantial relative improvement in MAE for the hardest classes (C3 and C4) when optimized for QWK. For Class 4 (10.4% of data), SAGE achieved a significant relative MAE reduction of 33.72% (from 2.565 MAE under CE (Acc) to 1.700 MAE under CE (QWK)). This improvement was even greater when SAGE used the DWCE loss and QWK selection, yielding a relative MAE reduction of 34.87% (2.676 MAE to 1.743 MAE), confirming that Accuracy optimization utterly masked severe errors in SAGE's ability to learn the traffic density distribution. The FSGCN model, which showed better performance, also demonstrated that DWCE coupled with QWK optimization yielded the best safety: for Class 3, FSGCN achieved a 18.7% relative reduction in MAE compared to its CE (Acc) baseline (1.797 MAE to 1.461 MAE). GCN also showed strong MAE reductions exceeding 18% for Class 3.
> > > >
> > > >
> > > >
> > > >
> > > > ### **Detailed results when target class is predicted wrong.**
> > > >
> > > > #### **A) Cost analysis on AMAZON-RATINGS**
> > > > **Metric:** Average Mean Absolute Error when the target class is predicted incorrectly.
> > > >
> > > > ### **FSGCN**
> > > > | Class | CE (Acc) | CE (QWK) | DWCE (Acc) | DWCE (QWK) |
> > > > |-------|-----------|-----------|-------------|--------------|
> > > > | 0     | 1.105±0.015 | 1.162±0.016 | 1.107±0.012 | 1.170±0.017 |
> > > > | 1     | 1.031±0.006 | 1.033±0.007 | 1.031±0.007 | 1.035±0.009 |
> > > > | 2     | 1.115±0.012 | 1.136±0.017 | 1.118±0.019 | 1.106±0.011 |
> > > > | 3     | 1.575±0.042 | 1.490±0.028 | 1.578±0.033 | 1.472±0.039 |
> > > > | 4     | 2.404±0.035 | 2.296±0.060 | 2.386±0.040 | 2.260±0.049 |
> > > >
> > > > ### **GCN**
> > > > | Class | CE (Acc) | CE (QWK) | DWCE (Acc) | DWCE (QWK) |
> > > > |-------|-----------|-----------|-------------|--------------|
> > > > | 0     | 1.273±0.037 | 1.317±0.027 | 1.262±0.033 | 1.295±0.039 |
> > > > | 1     | 1.082±0.020 | 1.119±0.024 | 1.084±0.018 | 1.107±0.020 |
> > > > | 2     | 1.332±0.050 | 1.232±0.032 | 1.298±0.048 | 1.222±0.036 |
> > > > | 3     | 1.853±0.048 | 1.625±0.059 | 1.826±0.065 | 1.611±0.054 |
> > > > | 4     | 2.764±0.087 | 2.225±0.056 | 2.685±0.044 | 2.259±0.074 |
> > > >
> > > > ### **SAGE**
> > > > | Class | CE (Acc) | CE (QWK) | DWCE (Acc) | DWCE (QWK) |
> > > > |-------|-----------|-----------|-------------|--------------|
> > > > | 0     | 1.178±0.019 | 1.335±0.031 | 1.173±0.025 | 1.321±0.037 |
> > > > | 1     | 1.047±0.008 | 1.106±0.017 | 1.051±0.007 | 1.117±0.024 |
> > > > | 2     | 1.222±0.027 | 1.271±0.035 | 1.226±0.018 | 1.256±0.030 |
> > > > | 3     | 1.802±0.039 | 1.694±0.064 | 1.821±0.038 | 1.697±0.058 |
> > > > | 4     | 2.701±0.050 | 2.400±0.069 | 2.729±0.079 | 2.463±0.105 |
> > > >
> > > > ### **GAT**
> > > > | Class | CE (Acc) | CE (QWK) | DWCE (Acc) | DWCE (QWK) |
> > > > |-------|-----------|-----------|-------------|--------------|
> > > > | 0     | 1.189±0.024 | 1.315±0.042 | 1.192±0.018 | 1.298±0.067 |
> > > > | 1     | 1.061±0.017 | 1.107±0.018 | 1.064±0.012 | 1.106±0.020 |
> > > > | 2     | 1.222±0.023 | 1.242±0.034 | 1.219±0.026 | 1.222±0.038 |
> > > > | 3     | 1.809±0.051 | 1.694±0.063 | 1.791±0.033 | 1.675±0.078 |
> > > > | 4     | 2.693±0.065 | 2.408±0.103 | 2.680±0.067 | 2.404±0.061 |
> > > >
> > > >
> > > > Continued

---

> > > > > ### Author Response · Authors · 2025-12-03
> > > > > **Continued**
> > > > >
> > > > > ### **B) Cost analysis on Squirrel-Filtered**
> > > > > **Metric:** Average Mean Absolute Error when the target class is predicted incorrectly.
> > > > >
> > > > > ### **FSGCN**
> > > > > | Class | CE (Acc)     | CE (QWK)     | DWCE (Acc)   | DWCE (QWK)   |
> > > > > |-------|--------------|--------------|--------------|--------------|
> > > > > | 0     | 1.533±0.213  | 1.440±0.100  | 1.618±0.196  | 1.437±0.086  |
> > > > > | 1     | 1.216±0.061  | 1.288±0.085  | 1.252±0.058  | 1.247±0.063  |
> > > > > | 2     | 1.374±0.147  | 1.190±0.032  | 1.400±0.121  | 1.224±0.041  |
> > > > > | 3     | 1.797±0.187  | 1.539±0.112  | 1.833±0.246  | 1.461±0.080  |
> > > > > | 4     | 1.887±0.218  | 1.585±0.097  | 1.880±0.402  | 1.517±0.130  |
> > > > >
> > > > > ### **GCN**
> > > > > | Class | CE (Acc)     | CE (QWK)     | DWCE (Acc)   | DWCE (QWK)   |
> > > > > |-------|--------------|--------------|--------------|--------------|
> > > > > | 0     | 1.563±0.142  | 1.322±0.077  | 1.451±0.182  | 1.311±0.106  |
> > > > > | 1     | 1.252±0.099  | 1.272±0.091  | 1.189±0.081  | 1.230±0.075  |
> > > > > | 2     | 1.412±0.047  | 1.186±0.057  | 1.406±0.136  | 1.220±0.068  |
> > > > > | 3     | 1.662±0.190  | 1.352±0.101  | 1.713±0.193  | 1.422±0.088  |
> > > > > | 4     | 1.675±0.209  | 1.486±0.124  | 1.649±0.167  | 1.530±0.116  |
> > > > >
> > > > > ### **SAGE**
> > > > > | Class | CE (Acc)     | CE (QWK)     | DWCE (Acc)   | DWCE (QWK)   |
> > > > > |-------|--------------|--------------|--------------|--------------|
> > > > > | 0     | 1.611±0.229  | 1.286±0.083  | 1.476±0.230  | 1.248±0.061  |
> > > > > | 1     | 1.075±0.068  | 1.178±0.073  | 1.030±0.019  | 1.178±0.085  |
> > > > > | 2     | 1.695±0.109  | 1.130±0.081  | 1.727±0.089  | 1.082±0.046  |
> > > > > | 3     | 2.148±0.213  | 1.502±0.107  | 2.293±0.109  | 1.523±0.107  |
> > > > > | 4     | 2.565±0.233  | 1.700±0.142  | 2.676±0.229  | 1.743±0.128  |
> > > > >
> > > > > ### **GAT**
> > > > > | Class | CE (Acc)     | CE (QWK)     | DWCE (Acc)   | DWCE (QWK)   |
> > > > > |-------|--------------|--------------|--------------|--------------|
> > > > > | 0     | 1.377±0.120  | 1.440±0.143  | 1.368±0.127  | 1.332±0.049  |
> > > > > | 1     | 1.131±0.075  | 1.147±0.091  | 1.085±0.060  | 1.127±0.041  |
> > > > > | 2     | 1.382±0.057  | 1.265±0.090  | 1.378±0.101  | 1.298±0.065  |
> > > > > | 3     | 1.736±0.122  | 1.452±0.138  | 1.732±0.079  | 1.608±0.108  |
> > > > > | 4     | 1.810±0.196  | 1.595±0.183  | 1.903±0.172  | 1.661±0.130  |
> > > > >
> > > > >
> > > > > **Conclusion:** These results collectively underscore that for ordinal tasks like Squirrel-filtered and Amazon-Ratings, relying solely on accuracy masks severe errors. Our analysis shows that incorporating metrics such as Cost of Mistakes (MAE on failure) and QWK is essential for reliable model selection, as it distinguishes between minor misclassifications and catastrophic failures that standard accuracy treats identically.
> > > > >
> > > > > We again thank the reviewer for suggesting these experiments, and highlighting the ordinal nature of these datasets which were ignored by the community. This allowed to further improve our work.

---

> ### Author Response · Authors · 2025-12-03
>
> **Q. Which metrics are appropriate for which datasets/settings and why:**
>
> We thank the reviewer for this constructive critique and agree that the paper must move beyond diagnosis to provide actionable guidance on which metrics are appropriate in which settings. Please find the response below.
>
>
> **1. Retrieval & High-Stakes Tasks (e.g., ogbg-molhiv):** For tasks where false positives are costly (e.g., drug discovery), we propose also studying utility metrics (NDCG@K, Precision@K) apart from AUROC. Our evaluation suggests inversion such as while a GCN model selected via standard AUROC achieved the highest score (0.7844), it performed poorly on the downstream top-k metric (NDCG@1%: 0.3232). Conversely, a SAGE model selected via our proposed AUCPR metric—despite a lower global ROC score—yielded significantly higher utility in the top 1% (NDCG@1%: 0.6011). This trend holds across different K values and precision metrics. Therefore, for retrieval-heavy tasks, we recommend using AUCPR for early stopping, as it correlates better with the top-ranking performance required in practice.
>
> **2. Ordinal Classification (e.g., Amazon-Ratings, Squirrel-filtered):** For datasets with intrinsic order (labels 1–5), we advocate for ordinal-aware metrics (QWK) alongside standard Accuracy. Accuracy treats all errors equivalently, penalizing "near misses" (e.g., predicting 4 stars for a 5-star item) the same as "catastrophic misses" (e.g., predicting 1 star). Our experiments reveal that high accuracy does not guarantee the model has learned the underlying class distribution. Reporting Quadratic Weighted Kappa (QWK) provides a more holistic evaluation aligned with the ordinal nature of the data.
>
>
>
>
> **Cost of ordinal mistakes:** Finally, we recommend that future studies quantify the severity of misclassifications by reporting the "Cost of Mistakes", eg:-MAE calculated on failure cases on ordinal datasets such as Amazon-Ratings and Squirrel-filtered. While accuracy measures hit-rate, it ignores how "wrong" a prediction is. Our analysis shows that optimizing for QWK significantly reduces the magnitude of errors. Hence, we argue that reporting the Cost of Mistakes provides critical insight into model reliability that accuracy alone cannot capture.
>
>
>
>
> **3. Fairness & Representation: Long-Tail Classification (e.g., ogbn-arxiv):** In practical settings such as classifying scientific literature for repository organization, datasets like *ogbn-arxiv* often exhibit significant skew, where a few dominant fields comprise nearly half the data while niche fields form a "long tail." In scenarios where diverse representation is a priority, relying solely on Accuracy can obscure poor performance on less frequent topics. Therefore, reporting metrics such as Balanced Accuracy(or classwise performances) offers a complementary view, ensuring the model maintains utility across the entire spectrum of subjects.
>
> We thank the reviewer for these suggestions, which improved our work.
>
>
>
>
>
>
> **Q. Squirrel-filtered original regression dataset:**
>
> We fully agree with the reviewer that predicting web traffic is inherently a regression task and could be reformulated as such by recovering the raw counts. However, the specific scope of our study is to audit the current GNN benchmarking ecosystem, which universally relies on the discretized 5-class version.
> To improve the current ecosystem, we treat it as Ordinal Regression (via QWK), a "middle ground." This approach respects the continuous nature of the underlying phenomenon by penalizing prediction distance—unlike standard accuracy—while maintaining compatibility with the dataset structure currently used by the community. This allows us to isolate and measure the impact of metric selection within the established experimental setup.

---

### Official Review · Reviewer_bwF3 · 2025-11-02

**Soundness:** 2
**Presentation:** 2
**Contribution:** 2
**Rating:** 2
**Confidence:** 4

**Summary:**

The authors identify several graph learning benchmarks that have significant class imbalance, yet are typically evaluated with class-imbalance-insensitive aggregate metrics like (standard) accuracy and AUROC, leading to diminished utility and potentially misleading conclusions on architectural effectiveness. They then demonstrate how class-imbalance-insensitive and -sensitive metrics and the models developed based on either sets of metrics differ through a series of studies.

**Strengths:**

**Stengths:**
1. The authors identify a very important and clear underlying issue in (a subset of) graph learning benchmarks.
2. The overall framing and message of the paper is very clear.
3. Most experimental studies, while not without limitations (See Weakness 3), are well set-up and demonstrate the effect of evaluation metric selection on the resulting models in a variety of scenarios.

**Weaknesses:**

**Weaknesses:**
1. There is little to no novelty in the paper; it reads more like a “position-track” paper, albeit with more empirical support. This is not to downplay the claims or potential impact of the paper — on the contrary, such papers on evaluation procedures are very valuable in steering graph learning (or ML in general) research in the right direction. However, the paper’s standalone contributions are very limited beyond (a) identifying graph learning datasets with class imbalance, and (b) arguing that they should use more class-imbalance-sensitive metrics, with experimental studies to compare evaluations using class-imbalance-sensitive vs. -insensitive metrics. The lack of novelty is also somewhat obscured by the fact that the paper does not refer to similar problems or studies in general ML research or specific ML domains _at all_, which is an issue in itself (see Weakness 2).
2. Class imbalance and appropriate metric usage has been fairly thoroughly discussed in general ML research and/or domain-specific research, which is not mentioned in related work. In fact, there are several counter-studies questioning the superiority of AUPRC in class imbalance settings [1, 2]; so while the overall argument for favoring AUPRC over AUROC is intuitive in certain cases or under some assumptions, I think the so-called superiority is not fully settled. I think the authors take this superiority for granted, and in the experimental section 4 simply show how the Acc vs Bal-Acc and AUROC vs AUPRC results (and downstream architectural decisions) differ, with little justification on why the results from one set of metrics are necessarily “better” than the other. While I do agree overall with the conclusions set forth by the authors, given the minimal novelty of the paper, I think the theoretical/empirical justifications provided need to be _much_ stronger than their current state to merit acceptance; something I expand upon in Weakness 3. Please also see the Questions section for some potential ideas towards this direction.
3. I think the “divergent metrics” and “optimizing for different objectives” studies in section 4.2 are decent initial attempts to justify the use of class-imbalance-sensitive metrics from an empirical standpoint. However, the conclusions fall quite flat because of Weakness 2. This leads to trivial concluding arguments such as:
   > In fig. 4, we observe that models selected based upon validation _balanced-accuracy_ yield a higher performance on _balanced-accuracy_, highlighting the choice of validation metric also plays a role in evaluation.

   It should come as a surprise to no one that “a model selected upon validation metric X yields a higher performance on the same metric X”. I think more effort is required to go beyond trivial statements (unless I am misunderstanding the point made here).

**Minor issues (no effect on score):**
1. Why call it _GNN_ benchmarks rather than _graph learning_ benchmarks? I think the term GNN is somewhat interchangably used either to refer to (a) most graph learning architectures in general, or (b) ones that specifically operate on the graph structure, the latter case being common when distinguishing models like graph transformers (GT) as a separate class of graph learning methods. While I think the meaning is clear in your case, I think referring to these datasets as _graph learning_ benchmarks is a straightforward solution that avoids any potential semantic ambiguity.
2. Typos:
   - Some opening parantheses (“(”) lack a preceding space across the paper.
   - L64-65: `\citet` used instead of `\citep` for Luo et al. (2024).

[1] Richardson, E., Trevizani, R., Greenbaum, J. A., Carter, H., Nielsen, M., & Peters, B. (2024). The receiver operating characteristic curve accurately assesses imbalanced datasets. Patterns (New York, N.Y.), 5(6), 100994. doi:10.1016/j.patter.2024.100994

[2] McDermott, M. B. A., Zhang, H., Hansen, L. H., Angelotti, G., & Gallifant, J. (2024). A closer look at AUROC and AUPRC under class imbalance. In Proceedings of the 38th International Conference on Neural Information Processing Systems (NeurIPS '24), Vol. 37. Curran Associates Inc., Red Hook, NY, USA, Article 1400, 44102–44163. doi:10.48550/ARXIV.2401.06091

**Questions:**

**Questions:**
1. (Suggestion) The authors need to lean further in on already existing analyses of relevant ML evaluation metrics (from both sides such as the aforementioned counter-studies), perhaps relying more on statistical studies and theoretical work (again, I think the counter-studies [1, 2] are good examples on how to approach this problem, independent of the conclusions that may contradict this work) rather than intuition to provide a more convincing theoretical footing and demonstrate a more tangible contribution for the paper.
2. (Suggestion) From a more empirical standpoint, the authors can look into whether models developed with the class-imbalance-sensitive metrics like AUPRC in the evaluation pipeline employ better generalization properties across tasks with different class distributions (e.g. in pre-training settings). I think providing several “case studies” in this vein that go beyond the examples associated with Weakness 3 would help demonstrate the merits of metric selection in a more clear manner.

**Conclusion:** I think this paper is significantly more suitable for a position paper track or a benchmarking track in its current state; the limited novelty and the inability to convincingly argue for its conclusions lead me to vote for a (somewhat harsh) clear reject; even though I very much like the premise and message of the paper. I certainly encourage the authors to revise the paper accordingly to arrive at a much stronger work.

---

> ### Author Response · Authors · 2025-12-03
> **Part 1**
>
> We appreciate the reviewer's recognition that this work identifies a very important underlying issue, and encouraged to find that they find the overall framing and message of the paper very clear. We below address the points raised by the reviewer and also conduct additional experiments.
>
>
> **Response to W1 and W2: ML class imbalance and AUCPR and AUROC debate**
>
> We acknowledge the reviewer’s valid point that class imbalance is a well-studied problem in general ML, and we apologize for the omission in the related work. Below we clarify the relationship between our work, general ML literature, and the specific debate surrounding AUPRC vs. AUROC and contextualize w.r.t the standard datasets from the graph learning literature that we have considered. We agree that explicitly connecting our graph-specific findings to general class imbalance strategies strengthens the work. In the final version, we will incorporate citations to established works (He & Garcia [3]), Rezvani et al.[9], Elkan et al.[6]), SMOTE[4]) to modern objectives like Focal Loss [8], and more works in this area. We again apologize for this omission, and we will definitely expand our related work on ML literature in this area.
>
> **Debate on AUROC vs. AUPRC:**
> Below we discuss the arguments for AUROC and AUPRC.
>
> We appreciate the valid critique regarding metric selection. We are aware of recent studies, such as Richardson et al. [1] and McDermott et al. [2], which argue that AUPRC is not a perfect metric due to baseline dependence and interpolation issues. We highlight that AUROC is often superior for context-independent evaluation where errors across the score spectrum must be treated uniformly[2].
>
> **Contextual Justification for AUPRC:** However, the choice of metric must be dictated by the deployment context. McDermott et al. [2] rigorously distinguish between "context-independent evaluation" (favoring AUROC) and "high-cost, single-group intervention prioritization" (favoring AUPRC). In context such as "selection of candidate molecules" cited in [2], the cost of False Positives in the high-confidence region significantly outweighs the cost of False Negatives, as only the most promising candidates proceed to validation. In such high-stakes retrieval settings, AUROC can be suboptimal because its uniform treatment of errors can mask poor performance in the critical "top-k" region.
>
> We believe the discussion on advantages of each metric would further strengthen the paper. We thank the reviewer for pointing this out.
>
> **Novelty Aspect:** In the later part of the response, we have incorporated the suggestions made by the reviewer specifically w.r.t impact of choosing model via AUPRC and AUROC on downstream utility.
>
> [1] Richardson, E., Trevizani, R., Greenbaum, J. A., Carter, H., Nielsen, M., & Peters, B. (2024). The receiver operating characteristic curve accurately assesses imbalanced datasets. Patterns, 5(6), 100994.
>
> [2] McDermott, M. B. A., Zhang, H., Hansen, L. H., Angelotti, G., & Gallifant, J. (2024). A closer look at AUROC and AUPRC under class imbalance. NeurIPS, Vol. 37.
>
> [3] He, H., & Garcia, E. A. (2009). Learning from imbalanced data. IEEE Transactions on Knowledge and Data Engineering, 21(9), 1263-1284.
>
> [4] Chawla, N. V., Bowyer, K. W., Hall, L. O., & Kegelmeyer, W. P. (2002). SMOTE: synthetic minority over-sampling technique. Journal of Artificial Intelligence Research, 16, 321-357.
>
> [5] Davis, J., & Goadrich, M. (2006). The relationship between Precision-Recall and ROC curves. ICML, 233-240.
>
> [6] Elkan, C. (2001). The foundations of cost-sensitive learning. IJCAI, 17, 973-978.
>
> [7] Liu, X. Y., Wu, J., & Zhou, Z. H. (2009). Exploratory undersampling for class-imbalance learning. IEEE Transactions on Systems, Man, and Cybernetics, Part B (Cybernetics), 39(2), 539-550.
>
> [8] Lin, T. Y., Goyal, P., Girshick, R., He, K., & Dollár, P. (2017). Focal loss for dense object detection. ICCV, 2980-2988.
>
> [9] Rezvani, Salim, and Xizhao Wang. "A broad review on class imbalance learning techniques." Applied Soft Computing 143 (2023): 110415.

---

> ### Author Response · Authors · 2025-12-03
> **Part 2.1**
>
> ***AUROC and AUPRC: Impact on downstream utility. Top-K candidates***
>
> We thank the reviewer for this insightful suggestion. We agree that providing empirical evidence linking metric selection to generalization performance significantly strengthens the paper. To address this, we conducted specific "case studies" to determine whether models selected via **Validation AUPRC** generalize better to downstream ranking tasks than those selected via **Validation AUROC**.
>
> ### 1. Experimental Design & Metric Context
>
> We trained GNNs on two distinct datasets characterized by high class imbalance. For every model configuration of hyperparameter, we saved two versions of the same model configuration:
>
> 1.  **Model-AUROC:** The checkpoint achieving the highest AUROC on the validation set.
> 2.  **Model-AUPRC:** The checkpoint achieving the highest AUPRC on the validation set.
>
> We evaluated these models on the held-out Test set using **NDCG@K** and **Precision@K** Scores for  K ∈ {1%, 2%, 3%, 4%, 5%}.
>
>
> *Precision@K (Purity):* Measures the fraction of true hits within the top cutoff. This serves as a direct proxy for ensuring that the limited budget for downstream verification is not wasted on false positives.
>
> *NDCG@K (Ranking Quality):* Normalized discounted cumulative gain. Goes beyond binary hit-rates to evaluate the ordering of candidates. It penalizes models that place true hits lower in the priority list, ensuring the most confident predictions are also the most accurate.
>
> To quantify the performance gap, we report the improvement ratio:
>
> $ \mathrm{Ratio} = \frac{\mathrm{Score}(\mathrm{Model_{AUPRC}})}{\mathrm{Score}(\mathrm{Model_{AUROC}})} $
>
> A ratio $>1$ indicates that tuning for AUPRC yields a superior model for deployment in terms of scores namely NDCG@K or PREC@K. We compute ratio for each configuration, and then report the mean and median ratio and p-value for different values of K.
>
> ---
>
> ### **1.1 Case Study A: Molecular Property Prediction (Molhiv)**
>
> **Context:** This is a graph classification task from the OGB benchmark, aiming to predict if a molecule inhibits HIV replication.
>
> **The Imbalance Challenge:** Active molecules are rare(~4%). In drug discovery pipelines ("virtual screening"), only the top few candidates are physically tested in a lab.
>
> **Why Metric Selection Matters:** A model must have high **Precision@Top-K**. False positives at the top of the list result in wasted laboratory resources.
>
> **Results:** Selecting checkpoints via AUPRC resulted in significant gains in Top-K utility across all architectures. For each of the models we observed that models selected via AUPRC obtained significantly higher Scores(Ratio >1) for NDCG@K and PREC@K for different thresholds. Further, the results are significant based upon the p-values obtained.
>
>
>
> ### **Model: GCN (Graph Convolutional Network)**
>
> | Metric/Score     | Top-K | Mean Ratio | Median Ratio | P-Value       |
> |------------|-------|--|--------------|---------------|
> |  **NDCG**          | 1%    | 1.243 | 1.081  | 2.56e-10      |
> |   | 2%    | 1.121    | 1.070    | 7.20e-12      |
> |     | 3%    | 1.095     | 1.063  | 2.13e-11      |
> |      | 4%    | 1.078     | 1.053  | 3.35e-10      |
> |      | 5%    | 1.071     | 1.055   | 4.37e-10      |
> | **PRECISION**| 1%    | 1.202     | 1.085      | 1.29e-11      |
> |            | 2%    | 1.091   | 1.067    | 5.35e-13      |
> |            | 3%    | 1.068 | 1.056    | 1.97e-10      |
> |            | 4%    | 1.050     | 1.044     | 3.63e-08      |
> |            | 5%    | 1.044    | 1.045      | 4.69e-07      |
> ### **Model: GAT (Graph Attention Network)**
>
> | Metric/Score     | Top-K | Mean Ratio | Median Ratio | P-Value     |
> |------------|-------|------------|--------------|-------------|
> |     **NDCG**       | 1%    | 1.707    | 1.279 | 3.33e-12 |
> |      | 2%    | 1.247    | 1.164   | 4.34e-12 |
> |    | 3%    | 1.180   | 1.170   | 1.93e-11 |
> |   | 4%    | 1.154   | 1.140   | 5.01e-11 |
> |      | 5%    | 1.133    | 1.133   | 1.82e-10 |
> | **PRECISION**        | 1%    | 1.581     | 1.217  | 5.39e-12 |
> |     | 2%    | 1.165     | 1.114  | 1.38e-10 |
> |    | 3%    | 1.103     | 1.083   | 1.25e-07 |
> |     | 4%    | 1.080     | 1.077  | 2.13e-06 |
> |       | 5%    | 1.059     | 1.064   | 1.15e-04 |
>
>
> ### **Model: GraphSAGE**
>
> | Metric/Score      | Top-K | Mean Ratio | Median Ratio | P-Value   |
> |------------|-------|----------|---------|-----------|
> | **NDCG**      | 1%    | 1.294              | 1.140    | 2.07e-09  |
> |    | 2%    | 1.148       | 1.095    | 2.90e-09  |
> |       | 3%    | 1.119       | 1.074     | 1.91e-09  |
> |      | 4%    | 1.108        | 1.070     | 5.23e-10  |
> |     | 5%    | 1.094        | 1.065         | 1.52e-09  |
> | **PRECISION** | 1%    | 1.214               | 1.115     | 9.61e-09  |
> |      | 2%    | 1.089      | 1.081     | 3.64e-07  |
> |       | 3%    | 1.065        | 1.045     | 9.06e-06  |
> |        | 4%    | 1.056 | 1.041     | 6.23e-06  |
> |      | 5%    | 1.041         | 1.021      | 1.71e-04  |

---

> ### Author Response · Authors · 2025-12-03
> **Part 2.2**
>
> ---
>
> ### **1.2 Case Study B:  Questions Dataset**
>
> **Context:** The Questions dataset is sourced from the Yandex Q community. It is a node classification task where the goal is to predict the status of users (nodes) based on their interaction graph.
>
> **The Imbalance Challenge:** The dataset is highly imbalanced, with ~97% of users belonging to the "active" class and ~3% being "deleted/blocked."
>
> **Results:**
> Similar to molhiv, the empirical results on this dataset also show that models optimized for AUPRC are significantly better at ranking the minority class at the top of the list (Top-K). The p-values are extremely low, conveying statistical significance.
>
> ### **Model: GAT (Graph Attention Network)**
>
> | Metric/Score      | Top-K | Mean Ratio | Median Ratio | P-Value      |
> |------------|-------|----------------------|-----------------------|-------------|
> | **NDCG**    | 1%    | 1.102               | 1.060              | 1.96e-32    |
> |             | 2%    | 1.078               | 1.058             | 1.08e-29    |
> |             | 3%    | 1.068               | 1.049               | 5.47e-27    |
> |             | 4%    | 1.060           | 1.040               | 7.02e-23    |
> |             | 5%    | 1.053           | 1.031               | 3.56e-18    |
> | **PRECISION**| 1%    | 1.092           | 1.061               | 1.74e-23    |
> |             | 2%    | 1.063          | 1.044              | 1.52e-17    |
> |             | 3%    | 1.051        | 1.035        | 5.84e-12    |
> |             | 4%    | 1.041          | 1.023         | 2.58e-07    |
> |             | 5%    | 1.032         | 1.020            | 1.04e-04    |
>
>
>
>
> ### **Model: GCN (Graph Convolutional Network)**
>
> | Metric/Score    | Top-K | Mean Ratio  | Median Ratio | P-Value      |
> |-----------|-------|----------------------|-----------------------|-------------|
> | **NDCG**| 1%    | 1.111        | 1.054           | 3.02e-41    |
> |             | 2%    | 1.073         | 1.042            | 5.94e-32    |
> |             | 3%    | 1.058       | 1.037         | 6.69e-29    |
> |             | 4%    | 1.048    | 1.031         | 1.16e-20    |
> |             | 5%    | 1.039       | 1.024         | 1.44e-14    |
> | **PRECISION** | 1%    | 1.095        | 1.051        | 2.96e-32    |
> |             | 2%    | 1.047      | 1.029        | 1.03e-12    |
> |             | 3%    | 1.030        | 1.023     | 2.65e-07    |
> |             | 4%    | 1.018          | 1.011         | 1.10e-02    |
> |             | 5%    | 1.008         | 1.000         | 0.781       |
>
>
>
> ### **Model: GraphSAGE**
>
> | Metric/Score     | Top-K | Mean Ratio  | Median Ratio  | P-Value      |
> |------------|-------|----------------------|-----------------------|-------------|
> | **NDCG**         | 1%    | 1.056       | 1.037          | 8.55e-24    |
> |             | 2%    | 1.048          | 1.035          | 2.02e-23    |
> |             | 3%    | 1.044          | 1.032           | 1.44e-19    |
> |             | 4%    | 1.038       | 1.028           | 8.43e-16    |
> |             | 5%    | 1.034          | 1.026          | 1.28e-14    |
> | **PRECISION** | 1%    | 1.055           | 1.036           | 3.13e-18    |
> |             | 2%    | 1.043           | 1.030             | 1.22e-15    |
> |             | 3%    | 1.038           | 1.027          | 2.08e-11    |
> |             | 4%    | 1.028           | 1.022           | 1.05e-06    |
> |             | 5%    | 1.022           | 1.021          | 3.53e-05    |
>
>
>
> ---
>
> ## 4. Conclusion
> We thank the reviewer for acknowledging the strengths of our paper, and also pushing us to strengthen the empirical justification of our claims, as our expanded investigation demonstrates that the choice between AUROC and AUPRC is a determining factor in downstream generalization utility rather than a mere preference. While we acknowledge recent literature preferring AUROC for context-independent evaluation, our results in lines with the analysis of McDermott et al.[1] confirm that validating with AUPRC leads to superior performance on top-k retrieval. Specifically, case studies on ogbg-molhiv and Questions reveal that models selected via AUPRC significantly outperform those selected via AUROC in NDCG@K and Precision@K metrics. This demonstrates that AUPRC-optimized models successfully prioritize true positives—a critical requirement where verification budgets are limited. Crucially, while the necessity of AUPRC is recognized for severely imbalanced datasets like ogbg-molpcba (~1.4%), we contend for datasets, specifically, ogbg-molhiv’s ~4% positive rate is also far from balanced; defaulting to AUROC here ignores the prohibitive cost of false positives in domains like drug discovery and could steer the research toward suboptimal models.
>
> We again thank the reviewer for the suggestion, which further improved our work.
>
>
> [1] McDermott, M. B. A., Zhang, H., Hansen, L. H., Angelotti, G., & Gallifant, J. (2024). A closer look at AUROC and AUPRC under class imbalance. NeurIPS, Vol. 37.

---

> > ### Author Response · Authors · 2025-12-03
> >
> > **Why call it GNN benchmarks rather than graph learning benchmarks?**
> >
> > We thank the reviewer for this suggestion, we agree to change the title accordingly.

---

### Author Response · Authors · 2025-12-03
**Summary of the rebuttal**

We thank the reviewers for their insights and constructive feedback. A comprehensive point-by-point response to the reviewers' comments is provided after each review. The major additional experiments are listed below.

1. AUCPR vs AUROC in model selection: For tasks like drug discovery(ogbg-molhiv), when there is a high cost of false positives, we compared the performance of models chosen by AUROC vs AUCPR and showed how top-k retrieval rankings(such as ndcg@k and prec@k) can significantly change and why it matters.

2. Ordinal Metrics for datasets which have an ordinal nature: We showed that commonly used standard accuracy fails to capture error severity in commonly used ordinal graph datasets (e.g., Amazon-ratings and Squirrel-filtered), and how optimizing for ordinality-aware losses and metrics can reduce the "Cost of Mistakes".

3. Rank Inversion: We provided empirical evidence of "rank inversions," demonstrating that the model identified as optimal by standard metrics (such as AUROC) can be suboptimal for downstream utility.

4. Actionable Evaluation Framework: We propose guidelines for evaluating on different datasets that take into account the nature of the datasets, the nature of the relationship between classes, along with loss functions for robust training.

We hope these revisions will satisfactorily address the concerns raised by the reviewers and elevate the overall quality of our work.

We apologize for the delay in our response, which was due to the time required to execute the additional experiments.

---

### Meta-Review · Area_Chair_7XXV · 2026-01-06

**Summary:**

The paper addresses an important problem in machine learning: selecting an appropriate evaluation metric. Specifically, the authors focus on graph machine learning and imbalanced datasets.

All the reviewers acknowledge that the problem addressed in this research is important and is often neglected. Thus, the paper may make a positive impact by motivating the community to use more appropriate metrics.

However, several limitations are mentioned by the reviewers (see the detailed list of limitations below). In my opinion, an important concern that remains unaddressed is that there is not much discussion about the considered measures and their properties and very limited discussion of related work. Indeed, in general machine learning literature, there are numerous papers discussing the measures from this study. Some of these papers are empirical, while others are theoretical and give important insights into, e.g., properties of the measures or key differences between some of them. The current research states that some measures are better than others for imbalanced datasets without going into much detail.

I appreciate the efforts the authors put into their rebuttal. There are additional experiments, guidelines, and discussions motivated by the reviewers’ comments. However, these new results are not incorporated in the paper, and it seems that to properly do this, the text may require noticeable changes and additional reviewing.

While I agree that the addressed topic is important, I suggest rejection at this stage, and I encourage the authors to incorporate the suggested updates. I believe that it may significantly strengthen the paper and increase its potential impact.

**Reviewer Concerns:**

The following concerns are mentioned by the reviewers:

1. Limited novelty, some observations are trivial (bwF3, vz42, TdYK, xAoy).
In the rebuttal, some new experiments are conducted, which partially resolve the issue.

2. Limited discussions of related work (bwF3).
The authors promised to address this issue in the final version.

3. Not clear why alternative evaluation measures are better; there is not much discussion about the measures (bwF3, vz42, TdYK).
The authors provided some discussions about that in the rebuttal. Personally, I believe that extended discussions about the properties of different measures and related work can be a significant part of the updated paper.

4. Narrow list of datasets (TdYK).
The authors explain their choice, and they also plan to add some new datasets to their analysis. Personally, I believe that considering reliable datasets is critical for this study, as its goal is to stimulate better evaluation practices in the graph ML field, and using reliable datasets is critical for proper evaluation.

5. The paper is more suitable for a position paper track (bwF3, xAoy).

6. No guidelines for future studies (xAoy).
The authors provided discussions about that in the rebuttal.

Overall, there are many new results provided in the rebuttal, but, as far as I understand, many of them are not incorporated in the text of the paper. In my opinion, the updated paper may change significantly from the originally submitted version and may also require additional reviews.

**Reviewer Scores:**

The initial review score are 2 (bwF3); 4 (vz42); 6 (TdYK); 2 (xAoy). In my opinion, Reviewer TdYK may increase their score based on the rebuttal. Some other scores can also potentially be increased but I believe these would remain below borderline.

---

### Decision · Program_Chairs · 2026-01-26

Reject